# Robust Signal Enhancement via Fractional Detail Views and Knowledge Guided Multi-view Fusion

**Zikun Jin** [1 2]  **Yuhua Qian** [1 2 3]  **Xinyan Liang** [1 2 3]  **Jiaqian Zhang** [1 2]  **Haijun Geng** [4]

## Abstract

Robust signal enhancement at low SNR is fundamentally challenging because noise becomes strongly entangled with the signal and corrupts local time–frequency (TF) evidence. In this regime, fixed resolution short time Fourier transform (STFT) enhancement with purely data driven convolutional biases can become overconfident in unreliable TF regions, causing unstable suppression or residual artifacts. We propose FracKGMF, which couples Fractional Distance Decay Convolution (FracConv) with Knowledge Guided Multi-view Fusion (KGMF) for expressive TF modeling and reliability aware decisions under heavy corruption. FracConv introduces a lightweight fractional distance decay family that reshapes local interactions into long tailed receptive patterns, enabling aggregation of weak but globally consistent cues when per-bin observations are ambiguous. KGMF uses a wiener inspired reliability prior to calibrate multi-view fusion and reduce excessive suppression in uncertain regions. Experiments on speech and EM benchmarks show consistent improvements over state-of-the-art baselines, with particularly large gains under extremely low SNR, including a 33 dB average improvement on EM signals at -20 dB.

## 1. Introduction

Recovering clean signals from interference dominated at extremely low SNR is a central challenge in speech enhancement and electromagnetic (EM) sensing (Défossez

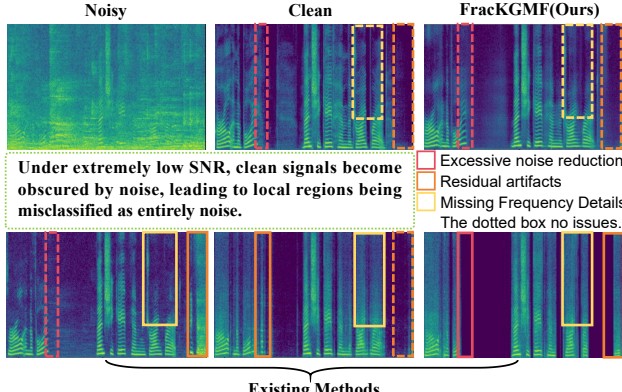

*Figure 1.* At extremely low SNR, clean structures are easily misclassified as noise, causing artifacts or excessive suppression in prior methods; FracKGMF better preserves signal patterns while suppressing interference.

et al., 2020; Su et al., 2023; Cheng & Wu, 2025). In this regime, noise is not a small additive perturbation: it becomes strongly coupled with the underlying signal, making local time-frequency (TF) evidence ambiguous and often misleading. Consequently, enhancement systems may become overconfident in corrupted TF regions, leading to unstable suppression, residual interference, and brittle behavior under changes in interference structure.

A large number of modern methods operate on the STFT spectrum (Wang et al., 2023; Mamun & Hansen, 2024; Wang et al., 2024), where informative structure may exist but is weak: harmonic ridges, narrowband carriers, and transient signatures are frequently submerged and partially aliased by interference (Zhang et al., 2024a;b). At low SNR, the key difficulty is not merely representing TF patterns but deciding what to trust when the observation is interference dominated. In Figure 1, existing methods (Défossez et al., 2020; Tai et al., 2023; Yang et al., 2025a) tend to residual artifacts, excessive noise reduction at low SNRs and missing frequency detail. Purely local convolutional inductive biases can fail because they aggregate evidence over a fixed, short range that is insufficient to separate weak targets from coupled noise; meanwhile, highly flexible unconstrained filters or implicit fusion can fit spurious interference variations, yielding unstable reconstructions and poor robustness ex-

[1]Institute of Big Data Science and Industry, Shanxi University, Taiyuan, China [2]Key Laboratory of Evolutionary Science Intelligence of Shanxi Province, Shanxi University, Taiyuan, China [3]School of Artificial Intelligence, Shanxi University, Datong, China. [4]School of Automation and Software Engineering, Shanxi University, Taiyuan, China. Correspondence to: Yuhua Qian <jinchengqyh@126.com>.

*Proceedings of the 43rd International Conference on Machine Learning*, Seoul, South Korea. PMLR 306, 2026. Copyright 2026 by the author(s).

actly when the noise coupling is strongest (Shin et al., 2023; P-J et al., 2023; Sun et al., 2024; Guo et al., 2024b). Recent diffusion and flow based approaches have further improved perceptual quality, but can be computationally intensive and may still exhibit instability when the conditioning signal is highly ambiguous (Strauss & Edler, 2021; Lu et al., 2022; Tai et al., 2023; Gonzalez et al., 2024).

This paper targets the above failure mode. We posit a coherent principle for robust enhancement under extremely low SNR: (i) use FracConv to construct a fractional detail (FD) view that profiles distance decay interactions on a local footprint for structure recovery, (ii) use the large receptive field (LRF) view as a long context, low variance representation that aggregates weak but spatially consistent TF evidence to stabilize ambiguous regions, and (iii) fuse the two views with an explicit reliability mechanism that calibrates per-bin weighting and prevents overreliance on any failing view.

First, FracConv is designed to capture dynamic TF details under extremely noise-signal coupling, rather than relying on unconstrained kernel shape discovery. It parameterizes local TF interactions with a family of fractional distance decay envelopes that continuously modulate how selectively the model responds to fine scale variations, while preserving directional sensitivity in the TF plane. By mixing multiple fractional decay index, FracConv adapts its effective correlation range across regions and noise levels, enabling it to accumulate weak but persistent signal cues that remain consistent even when per-bin observations are heavily corrupted. This structured, low dimensional design targets the low SNR coupling regime where unconstrained filters tend to chase interference variability and collapse out of distribution.

Second, under heavy interference, complementary views can fail in different TF regions; KGMF introduces reliability aware fusion for stable enhancement. It uses a Wiener inspired confidence signal derived from the noisy STFT as a physically grounded reference to calibrate fusion, discouraging overconfident reliance on a single view in interference dominated regions while allowing aggressive restoration when confidence is high. In short, FracConv creates a robust, structured complementary view; KGMF ensures that the system uses it safely under coupling, rather than trusting whichever view.

Empirically, FracKGMF consistently improves performance on speech and realistic EM benchmarks, especially in the extremely low SNR regime. These results are supported by theory showing that (i) FracConv provides expressive yet stable feature extraction via structured fractional decay with controlled operator magnitude, and (ii) KGMF yields reliability aware fusion with provable risk control under Wiener guided weighting. In particular, on realistic EM at $-20$ dB, where noise-signal entanglement is most severe, FracKGMF achieves the largest gains over the strongest recent baselines, highlighting the value of structured interactions and reliability aware fusion. Contributions are:

- We identify noise-signal entanglement at extremely low SNR as the core failure mode for TF enhancement, motivating structured evidence aggregation plus explicit reliability control.

- We propose FracKGMF, a unified framework where FracConv (structured fractional distance decay interactions) and KGMF (Wiener inspired reliability fusion) are jointly designed to prevent brittle enhancement under interference dominated observations.

- We validate consistent gains on speech and realistic EM benchmarks, highlighting particularly large improvements at -20 dB EM, demonstrating practical robustness in the most challenging coupling regime.

## 2. Related Work

**Time-frequency (TF) enhancement and robustness under low SNR.** TF representations such as the STFT are a standard front end for speech enhancement and are increasingly used in EM sensing (Wang et al., 2025), where neural models estimate complex spectra (Ishwarya & Kothandaraman, 2025) or TF masks with convolutional (Hu et al., 2020), recurrent, or attention based backbones (Parisae & Bhavanam, 2025). To improve robustness, many works expand contextual aggregation via deeper networks, larger kernels, dilation, or multi-scale designs to exploit long range continuity (e.g., harmonic persistence or carrier stability) (Sun et al., 2025). However, under extremely low SNR interference, local evidence is often unreliable and interference dominates observations, so simply expanding capacity or receptive field can amplify sensitivity to nuisance variations, producing artifacts and weak generalization (Richter et al., 2023; Fan et al., 2025).

**View construction and multi-view fusion.** Recent enhancement models increasingly go beyond a single STFT view by introducing auxiliary representations to better capture heterogeneous time-frequency structure (Xu & Zhang, 2025). Fractional domain representations, such as fractional Fourier analysis and its short time variants, provide a principled way to interpolate between time and frequency views, and have inspired auxiliary representations for handling heterogeneous TF patterns (Jin et al., 2025; 2026). In contrast to fixed index fractional transforms or fixed interaction shapes, our work focuses on learnable, low dimensional families of distance decay profiles and their stable integration into enhancement networks, enabling adaptive emphasis across TF regions without resorting to unconstrained kernel shape discovery. Multi-view and multi-branch fusion further aims to combine complementary failure modes (Liang et al.,

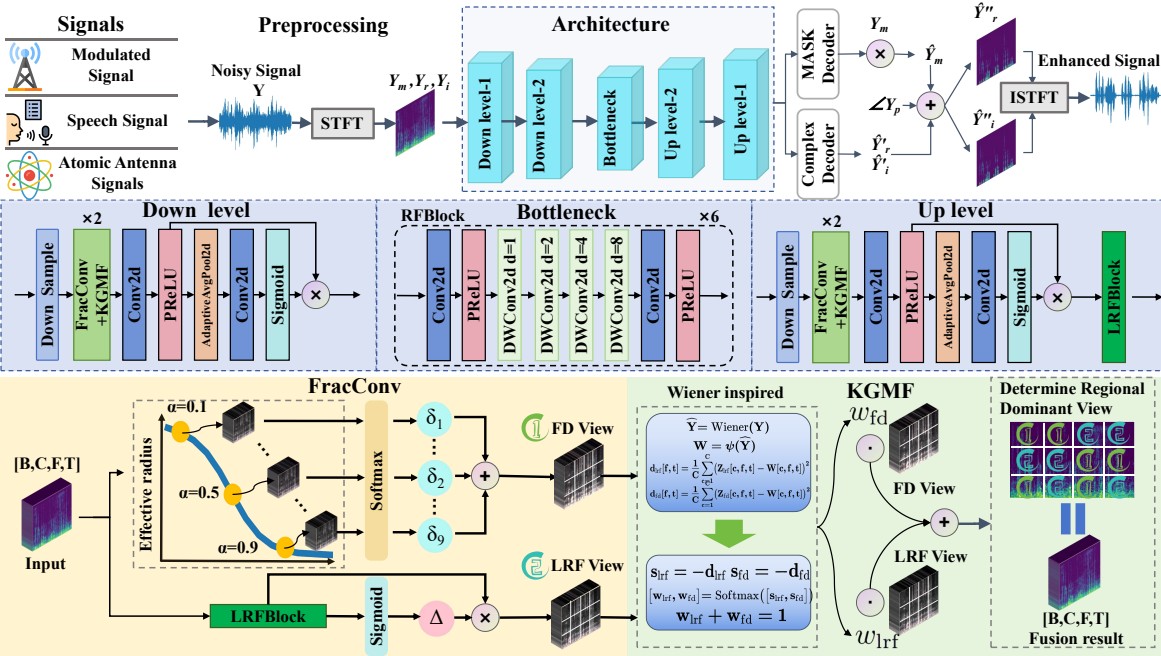

*Figure 2.* **Overview of FracKGMF.** We fuse LRF view with FD view under a Wiener inspired reliability prior for low SNR enhancement.

2022; Guo et al., 2024a; Yang et al., 2025b). Studies on multivariate one-dimensional data also highlight the importance of cross variable dependency modeling and correlation aware learning (Qiu et al., 2024; 2025a; Wu et al., 2025; Qiu et al., 2025b). However, in severely corrupted enhancement regimes, different branches may become overconfident in different TF regions, making reliability estimation essential (Liang et al., 2025). Classical priors such as Wiener filtering (Zhao & Zhang, 2025) provide physically grounded statistics that can act as a reliability reference when learning is uncertain. However, most existing methods (Xu & Zhang, 2023) either fix the view definition (e.g., a preset transform or interaction shape) or rely on implicit fusion, which can become miscalibrated and overly trust a failing branch when observations are interference dominated at extremely low SNR. In response, we construct a structured fractional detail view using a learnable, low dimensional family of distance decay profiles, and fuse it with statistics derived reliability guidance for stable enhancement in the low SNR regime.

## 3. Method

This section describes the design of FracKGMF (Figure 2). At extremely low SNR, overconfident decisions based on interference dominated TF evidence often lead to artifacts or excessive suppression. FracKGMF mitigates this failure mode by combining two views: an **Large Receptive Field (LRF) view** that stabilizes enhancement via context aggregation, and a **Fractional Detail (FD) View** that surfaces fine-grained yet persistent cues when per-bin observations

are ambiguous. These views are then fused in **KGMF** under a wiener reliability reference, which discourages overreliance on any failing view and enables stable restoration.

### 3.1. Large Receptive Field (LRF) View Construction

LRF view is a coverage oriented TF view that aggregates weak but spatially consistent cues over a large context to stabilize enhancement under interference dominated, extremely low SNR observations. It is implemented as an efficient large receptive field extractor that expands the contextual range while keeping the computation lightweight. Given $X \in \mathbb{R}^{C \times F \times T}$, the block stacks lightweight residual depthwise dilated convolution to deliver efficient long context coverage for accumulating weak yet consistent TF cues with low parameter and activation memory cost, without resorting to costly global attention. This stable view complements FracConv, which explicitly profiles distance decay interactions on a local footprint (Sec. 3.2), and the two are fused by KGMF using a wiener derived reliability reference (Sec. 3.4).

### 3.2. Fractional Detail View Construction: Profiling Distance Decay Interactions

We construct the FD view via **FracConv**, a structured depthwise operator that parameterizes distance decay interaction profiles on a fixed local footprint. Let the kernel size be $\mathbf{k} = (k_h, k_w)$ with center $c_y = (k_h - 1)/2$ and

$c_x = (k_w - 1)/2$. We define the kernel offset set

$$\Omega_{\mathbf{k}} = \big\{ u = (\Delta f, \Delta t) : \Delta f \in \big[ -\lfloor k_h/2 \rfloor, \lfloor k_h/2 \rfloor \big],$$
$$\Delta t \in \big[ -\lfloor k_w/2 \rfloor, \lfloor k_w/2 \rfloor \big] \big\}. \tag{1}$$

and assign each offset a radius $r(u) = \|u\|_2$.

### 3.2.1. FRACTIONAL RADIAL ENVELOPES AND MULTI-INDEX MIXING

For a decay index $\alpha > 0$, we define an unnormalized radial envelope

$$\widetilde{W}_\alpha(u) = (\varepsilon + r(u))^{-\alpha}, \qquad \varepsilon > 0, \tag{2}$$

and normalize it to obtain a bounded envelope on footprint:

$$W_\alpha(u) = \frac{\widetilde{W}_\alpha(u)}{\sum_{u' \in \Omega_{\mathbf{k}}} \widetilde{W}_\alpha(u')}, \qquad 0 \le W_\alpha(u) \le 1. \tag{3}$$

To adapt the decay profile with limited degrees of freedom, we use $M$ fixed indices $\{\alpha_m\}_{m=1}^M$ and learn a simplex mixture $\boldsymbol{w} \in \Delta^{M-1}$ via

$$\boldsymbol{w} = \mathrm{softmax}(\boldsymbol{\pi}), \qquad \boldsymbol{\pi} \in \mathbb{R}^M. \tag{4}$$

The resulting mixed envelope is

$$\Phi_{\boldsymbol{w}}(u) = \sum_{m=1}^M w_m W_{\alpha_m}(u), \tag{5}$$

which imposes a controlled radial distance decay profile while keeping the operator bounded and lightweight.

### 3.2.2. KERNEL FACTORIZATION AND INDUCED DEPTHWISE OPERATOR

For each depthwise channel $c$, let $A_c \in \mathbb{R}^{k_h \times k_w}$ be a learnable anisotropic base kernel. We define the effective kernel by factorization:

$$K_c(u) = A_c(u) \cdot \Phi_{\boldsymbol{w}}(u), \tag{6}$$

equivalently $\widehat{K}_c = A_c \odot \Phi_{\boldsymbol{w}}$. The induced fractional depthwise convolution is

$$\begin{aligned} (\mathrm{FracConv}(X))_c[i] &= \sum_{u \in \Omega_{\mathbf{k}}} K_c(u) X_c[i-u] \\ &= \mathrm{Conv2d}\big(X_c; \widehat{K}_c\big), \end{aligned} \tag{7}$$

where $\Phi_{\boldsymbol{w}}$ controls distance decay interactions and $A_c$ preserves directional selectivity. Compared with unconstrained kernel learning, the bounded envelope and convex mixing reduce estimator variance in interference dominated TF regions while retaining flexibility to capture fine structures.

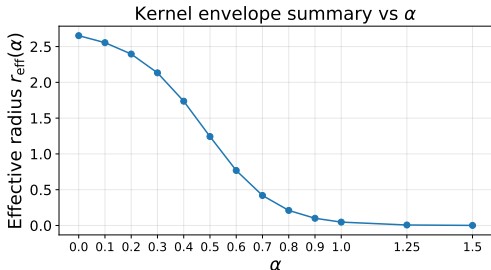

*Figure 3.* Envelope concentration vs. fractional decay indices $\alpha$, summarized by the effective radius $r_{\mathrm{eff}}(\alpha) = \sum_u W_\alpha(u) \|u\|_2$ on a fixed kernel footprint. The curve shows substantial variation over $\alpha \in [0.1, 0.9]$ and diminishing changes for larger $\alpha$, motivating our anchor range.

**Choice of fractional decay indices.** We use $M{=}9$ fixed indices $\{\alpha_m\}_{m=1}^9$ uniformly spaced in $[0.1, 0.9]$. Figure 3 visualizes how envelope concentration varies with $\alpha$ via an effective radius statistic. Most practically relevant shape variation occurs within $\alpha \in [0.1, 0.9]$, while larger $\alpha$ yields diminishing changes (increasingly center focused envelopes), making additional high-$\alpha$ anchors largely redundant. We also avoid $\alpha$ close to 0, where $(\varepsilon + r)^{-\alpha}$ becomes nearly constant and the envelope degenerates toward an almost uniform kernel, weakening the intended distance dependent inductive bias. Overall, the $[0.1, 0.9]$ anchor set provides a compact yet expressive discretization that supports stable simplex mixing to synthesize a continuum of interaction profiles with minimal overhead.

### 3.3. Wiener Prior as a Reliability Reference

We derive a Wiener inspired reliability prior from the noisy STFT $Y$ to provide a statistically grounded reference for fusion. Let $P_Y = |Y|^2$ be the noisy power spectrum. Using standard TF smoothing and noise tracking, we obtain the estimated signal and noise PSDs $(\widehat{P}_S, \widehat{P}_N)$ and define the per-bin recoverability as

$$R[f, t] = \frac{\widehat{P}_S[f, t]}{\widehat{P}_S[f, t] + \widehat{P}_N[f, t] + \epsilon_{\mathrm{w}}}, \qquad \epsilon_{\mathrm{w}} > 0. \tag{8}$$

Here, $R[f, t] \in [0, 1]$ becomes small in interference dominated regions and large in signal dominated regions. We do not use this estimate as the final denoised output; instead, it serves as a conservative TF reference for view selection and provides an explicit noise aware reliability cue for knowledge guided fusion.

### 3.4. Knowledge Guided Multi-view Fusion (KGMF)

KGMF fuses a LRF View with a FD View under the wiener derived reliability reference. Given an input TF feature map $X \in \mathbb{R}^{C_{\mathrm{in}} \times F \times T}$, we first align channels and extract two

parallel views:

$$H = \phi_{\text{in}}(X), \qquad Z_{\text{lrf}} = \mathcal{T}(H), \qquad Z_{\text{fd}} = \mathcal{F}(H), \quad (9)$$

where $\phi_{\text{in}}$ is a $1 \times 1$ projection or identity mapping, $\mathcal{T}$ is the LRF view extractor, and $\mathcal{F}$ is the FD view extractor. From the noisy complex STFT, we compute the wiener style TF estimate $\widehat{Y} = R \odot Y$, where $R$ is defined in Eq. (8), and project it into the same feature space:

$$W = \psi(\widehat{Y}), \qquad \psi : \mathbb{R}^{2 \times F \times T} \to \mathbb{R}^{C \times F \times T}. \quad (10)$$

The feature map $W$ provides a conservative reference for view selection. We then measure the consistency between each view and $W$ by channel averaged squared distance:

$$d_{\text{lrf}}[f, t] = \frac{1}{C} \sum_{c=1}^{C} \big(Z_{\text{lrf}}[c, f, t] - W[c, f, t]\big)^2,$$
$$d_{\text{fd}}[f, t] = \frac{1}{C} \sum_{c=1}^{C} \big(Z_{\text{fd}}[c, f, t] - W[c, f, t]\big)^2. \quad (11)$$

Using negative distances as scores, we compute per-bin softmax weights and fuse the two views:

$$[w_{\text{lrf}}, w_{\text{fd}}] = \text{Softmax}\big([-d_{\text{lrf}}, -d_{\text{fd}}]\big),$$
$$Z_{\text{mix}} = w_{\text{lrf}} \odot Z_{\text{lrf}} + w_{\text{fd}} \odot Z_{\text{fd}}. \quad (12)$$

Thus, the Wiener prior does not replace the learned views, but calibrates their per-bin contributions by favoring the view more consistent with the noise aware reference, reducing overconfident reliance on a failing view under severe corruption.

# 4. Theory and Design Justification

Our theory supports three design goals aligned with Sec. 3: (i) expressive yet structured distance decay interactions for the fractional view (Sec. 3.2); (ii) numerically stable gated of the fractional view; and (iii) robust knowledge guided fusion that learns a reliability aware view weighting using a wiener derived reference (Sec. 3.4). Proofs are provided in the appendix.

## 4.1. Expressivity via Mixtures of Fractional Envelopes

FracConv constructs a radial distance decay envelope on the finite kernel footprint $\Omega_{\mathbf{k}}$ by simplex mixing of a small set of fixed decay indices $\{\alpha_m\}_{m=1}^{M}$ (Eq. (5)). The following result formalizes that this discrete mixture can approximate a continuum of normalized fractional envelopes on $\Omega_{\mathbf{k}}$, with an $O(1/M)$ error.

**Theorem 4.1** (Approximation of FracConv distance decay envelopes)**.** *Let $\Omega_{\mathbf{k}}$ be the kernel offset set (Sec. 3.2) and $r(u) = \|u\|_2$ for $u \in \Omega_{\mathbf{k}}$. Fix $\varepsilon > 0$ and define*

$$W_\alpha(u) = \frac{(\varepsilon + r(u))^{-\alpha}}{\sum_{v \in \Omega_{\mathbf{k}}} (\varepsilon + r(v))^{-\alpha}}, \qquad \alpha \in [\alpha_{\min}, \alpha_{\max}]. \quad (13)$$

*Let $q$ be any mixing density on $[\alpha_{\min}, \alpha_{\max}]$ with $q(\alpha) \geq 0$ and $\int q(\alpha)\, d\alpha = 1$. Define*

$$\Phi_q(u) = \int_{\alpha_{\min}}^{\alpha_{\max}} q(\alpha) W_\alpha(u)\, d\alpha. \quad (14)$$

*Let*

$$\Delta = \frac{\alpha_{\max} - \alpha_{\min}}{M},$$
$$\alpha_m = \alpha_{\min} + \left(m - \frac{1}{2}\right)\Delta, \quad (15)$$
$$m = 1, \ldots, M.$$

*Define*

$$w_m = \int_{I_m} q(\alpha)\, d\alpha,$$
$$I_m = [\alpha_{\min} + (m-1)\Delta, \alpha_{\min} + m\Delta]. \quad (16)$$
$$m = 1, \ldots, M.$$

*Then*

$$\Phi_{\mathbf{w}}(u) = \sum_{m=1}^{M} w_m W_{\alpha_m}(u) \quad (17)$$

*satisfies*

$$\sup_{u \in \Omega_{\mathbf{k}}} \big|\Phi_q(u) - \Phi_{\mathbf{w}}(u)\big| \leq \frac{\alpha_{\max} - \alpha_{\min}}{2M} C_W, \quad (18)$$

*where*

$$C_W = \sup_{\alpha \in [\alpha_{\min}, \alpha_{\max}]} \sup_{u \in \Omega_{\mathbf{k}}} |\partial_\alpha W_\alpha(u)|. \quad (19)$$

*Moreover, $C_W$ admits an explicit finite bound as shown in Lemma A.1.*

**Remark.** If $w_m > 0$ for all $m$, then $\mathbf{w}$ can be represented exactly by softmax logits, e.g., $\pi_m = \log w_m + c$. If some $w_m = 0$, it can be approximated arbitrarily closely by finite logits. Thus, the approximation result is compatible with the softmax-parameterized mixture in Eq. (4).

## 4.2. Stability of Multi-index Mixing in FracConv

FracConv constructs the fractional distance decay envelope via simplex mixing $\Phi_{\mathbf{w}} = \sum_{m=1}^{M} w_m W_{\alpha_m}$ (Eq. (5)), and factorizes the effective depthwise kernel as $K_{\mathbf{w}} = A \odot \Phi_{\mathbf{w}}$ (Eq. (6)). Since $0 \leq W_{\alpha_m}(u) \leq 1$ on the finite footprint $\Omega_{\mathbf{k}}$ and $\mathbf{w} \in \Delta^{M-1}$, the mixed envelope remains bounded and does not increase the worst case gain beyond that of the base kernel, yielding a uniform stability guarantee. We assume standard discrete convolution with circular or zero padding, under which Young's inequality applies.

**Proposition 4.2** (Stability of simplex mixed FracConv)**.** *Consider one depthwise channel on footprint $\Omega_{\mathbf{k}}$ and define*

$$\Phi_{\mathbf{w}}[u] = \sum_{m=1}^{M} w_m\, W_{\alpha_m}(u), \qquad \mathbf{w} \in \Delta^{M-1},$$
$$K_{\mathbf{w}} = A \odot \Phi_{\mathbf{w}}, \quad (20)$$

where each $W_{\alpha_m}$ is the normalized envelope defined in Eq. (13).

Assume $\|A\|_1 \le B$ and let $F_{\boldsymbol{w}}(X) = K_{\boldsymbol{w}} * X$ be the induced convolution. Then for all $\boldsymbol{w} \in \Delta^{M-1}$,

$$\|F_{\boldsymbol{w}}(X)\|_2 \le B\|X\|_2. \qquad (21)$$

Moreover, for any $\boldsymbol{w}, \boldsymbol{w}' \in \Delta^{M-1}$,

$$\|F_{\boldsymbol{w}}(X) - F_{\boldsymbol{w}'}(X)\|_2 \le B\,\|\boldsymbol{w} - \boldsymbol{w}'\|_1\,\|X\|_2. \qquad (22)$$

**Corollary 4.3** (Envelope perturbation under weight changes). *For any* $\boldsymbol{w}, \boldsymbol{w}' \in \Delta^{M-1}$,

$$\|\Phi_{\boldsymbol{w}} - \Phi_{\boldsymbol{w}'}\|_\infty \le \|\boldsymbol{w} - \boldsymbol{w}'\|_1. \qquad (23)$$

**Implication.** Eq. (21) shows that multi-index mixing preserves a uniform gain bound, while Eq. (22) quantifies that the operator varies smoothly with the simplex weights.

## 4.3. Enhancement Driven Risk Control via Reliability Aware Fusion

KGMF improves enhancement quality by performing reliability aware fusion between a Large Receptive Field (LRF) View and a Fractional Detail (FD) View. Let $\widehat{S}_{\mathrm{lrf}}$ and $\widehat{S}_{\mathrm{fd}}$ denote the spectrum estimators produced by the LRF and FD branches, respectively. We consider the convex fusion estimator

$$\widehat{S}_\lambda = (1 - \lambda)\widehat{S}_{\mathrm{fd}} + \lambda\widehat{S}_{\mathrm{lrf}}, \qquad \lambda \in [0, 1], \qquad (24)$$

where $\lambda$ controls the contribution of the LRF view.

KGMF uses the wiener derived reference to construct perbin distance maps $d_{\mathrm{lrf}}$ and $d_{\mathrm{fd}}$ according to Eq. (11). The reliability statistic is defined as

$$Z := d_{\mathrm{fd}} - d_{\mathrm{lrf}}. \qquad (25)$$

For the two view softmax weighting in Eq. (12), the induced LRF fusion weight reduces to a logistic selector:

$$\widehat{\lambda}(Z) = w_{\mathrm{lrf}} = \frac{\exp(-d_{\mathrm{lrf}})}{\exp(-d_{\mathrm{lrf}}) + \exp(-d_{\mathrm{fd}})} = \sigma(Z), \qquad (26)$$
$$w_{\mathrm{fd}} = 1 - w_{\mathrm{lrf}}.$$

Thus, when the LRF view is closer to the wiener derived reference than the FD view, i.e., $d_{\mathrm{lrf}} < d_{\mathrm{fd}}$, the statistic $Z$ becomes positive and the fusion assigns a larger weight to the LRF estimator.

**Lemma 4.4** (Stability of the KGMF selector). *The KGMF selector* $\widehat{\lambda}(Z) = \sigma(Z)$ *is* $\frac{1}{4}$*-Lipschitz:*

$$|\widehat{\lambda}(Z) - \widehat{\lambda}(Z')| \le \frac{1}{4}\,|Z - Z'|. \qquad (27)$$

*For tensor-valued reliability statistics, the bound holds pointwise.*

Given the reliability statistic $Z$, define the conditional optimal fusion weight as

$$\lambda^\star(Z) = \arg\min_{\lambda \in [0,1]} \mathbb{E}\Big[\|\widehat{S}_\lambda - S\|_2^2 \,\Big|\, Z\Big]. \qquad (28)$$

This oracle weight characterizes the best possible convex fusion between the LRF and FD estimators under the conditional MSE criterion.

**Proposition 4.5** (Regret bound for reliability aware fusion). *Assume that the selected conditional minimizer* $\lambda^\star(Z)$ *satisfies the first-order optimality condition*

$$\mathbb{E}\Big[\big\langle \widehat{S}_{\lambda^\star} - S,\, \widehat{S}_{\mathrm{lrf}} - \widehat{S}_{\mathrm{fd}} \big\rangle \,\Big|\, Z\Big] = 0 \quad a.s. \qquad (29)$$

*This condition holds, for example, when* $\lambda^\star(Z)$ *is an interior minimizer. If the KGMF rule outputs* $\widehat{\lambda}(Z)$ *such that*

$$|\widehat{\lambda}(Z) - \lambda^\star(Z)| \le \varepsilon \quad a.s., \qquad (30)$$

*then*

$$\mathbb{E}\|\widehat{S}_{\widehat{\lambda}} - S\|_2^2 \le \mathbb{E}\|\widehat{S}_{\lambda^\star} - S\|_2^2 + \varepsilon^2\,\mathbb{E}\|\widehat{S}_{\mathrm{lrf}} - \widehat{S}_{\mathrm{fd}}\|_2^2, \qquad (31)$$

*where* $\widehat{S}_{\widehat{\lambda}} := \widehat{S}_\lambda\big|_{\lambda = \widehat{\lambda}(Z)}$.

*Moreover, suppose there exists an ideal reliability statistic* $\widetilde{Z}$ *such that*

$$|\widehat{\lambda}(\widetilde{Z}) - \lambda^\star(Z)| \le \varepsilon_0 \quad a.s. \qquad (32)$$

*If the observed statistic satisfies* $Z = \widetilde{Z} + \delta$ *and* $\widehat{\lambda}$ *is* $L_\lambda$*-Lipschitz in* $Z$, *then*

$$|\widehat{\lambda}(Z) - \lambda^\star(Z)| \le \varepsilon_0 + L_\lambda\|\delta\| \quad a.s. \qquad (33)$$

*In particular, Lemma 4.4 gives* $L_\lambda \le \frac{1}{4}$ *for the sigmoid KGMF selector.*

**Corollary 4.6** (Conditional optimal fusion is no worse than either view). *Conditioned on* $Z$, *the oracle fusion risk is no worse than using either single view:*

$$\mathbb{E}\Big[\|\widehat{S}_{\lambda^\star} - S\|_2^2 \,\Big|\, Z\Big] \le \min\Big\{\mathbb{E}\Big[\|\widehat{S}_{\mathrm{lrf}} - S\|_2^2 \,\Big|\, Z\Big],$$
$$\mathbb{E}\Big[\|\widehat{S}_{\mathrm{fd}} - S\|_2^2 \,\Big|\, Z\Big]\Big\}. \qquad (34)$$

*Therefore, when the wiener derived reliability statistic makes* $\widehat{\lambda}(Z)$ *close to* $\lambda^\star(Z)$, *KGMF approaches the conditional oracle fusion and yields an estimate closer to the clean target in expected MSE.*

**Remark (Why FracConv+KGMF can improve enhancement).** Eq. (31) decomposes the fusion risk into two components: (i) the conditional oracle risk $\mathbb{E}\|\widehat{S}_{\lambda^\star} - S\|_2^2$, which reflects the best achievable convex combination of the LRF and FD views, and (ii) the approximation gap between the

implemented fusion weight $\widehat{\lambda}(Z)$ and the conditional optimum $\lambda^{\star}(Z)$. FracConv is designed to strengthen the FD view $\widehat{S}_{\mathrm{fd}}$ by enforcing structured and bounded distance decay interactions, thereby reducing unstable interference fitting and providing a more reliable complementary estimator to the LRF view. Under this interpretation, better view complementarity lowers the oracle fusion risk, while the wiener derived reliability statistic helps the implemented selector track the target fusion weight. Together, these effects explain why FracConv+KGMF can produce an enhanced estimate that is closer to the clean target in expected MSE.

## 5. Experiments

This section evaluates **FracKGMF** under extremely low SNR noise-signal entanglement. We test three claims: (i) FracConv improves detail-sensitive feature extraction under interference dominated TF observations; (ii) KGMF provides reliable gains over single view and naive fusion via statistics-derived confidence; and (iii) FracKGMF remains effective across SNRs and conditions, with the strongest gains at the lowest SNRs. We report objective and perceptual metrics on speech and realistic EM benchmarks, and conduct controlled ablations for each component.

### 5.1. Signal Enhancement Datasets

We evaluated EM and speech enhancement on five datasets.

**EM datasets (4-bins/20-bins Atomic Antenna Signals).** We evaluate on the public multifrequency microwave datasets released in the Nature Communications study on deep learning enhanced Rydberg receivers (Liu et al., 2022). The datasets are built under an FDM-style setting with either 4 or 20 frequency bins, where each sample is a noisy spectrum/IQ observation and the goal is to recover the underlying signal information under strong interference and noise coupling. We follow the original data protocol and report results across a wide range of SNRs, focusing on the extremely low SNR regime (-20dB) where the task is most challenging.

**Modulation datasets.** We generate a RadioML-style modulation dataset following the DeepSig RML2018 protocol (O'Shea et al., 2018). Each sample is a length(48,000) real passband waveform with aligned noisy/clean pairs $(y, x)$. We simulate 12 carrier bands (0.02–29.51 GHz) and multiple modulations, and corrupt signals with the same four noise processes over SNR $\{-20, \ldots, 20\}$ dB.

**VoiceBank+DEMAND.** A canonical speech enhancement benchmark (Botinhao et al., 2016) with speaker disjoint train/test splits and controlled mixture generation, including SNR mismatch (train: 0/5/10/15 dB; test: 2.5/7.5/12.5/17.5 dB) to evaluate generalization under moderate noise level shifts (16 kHz).

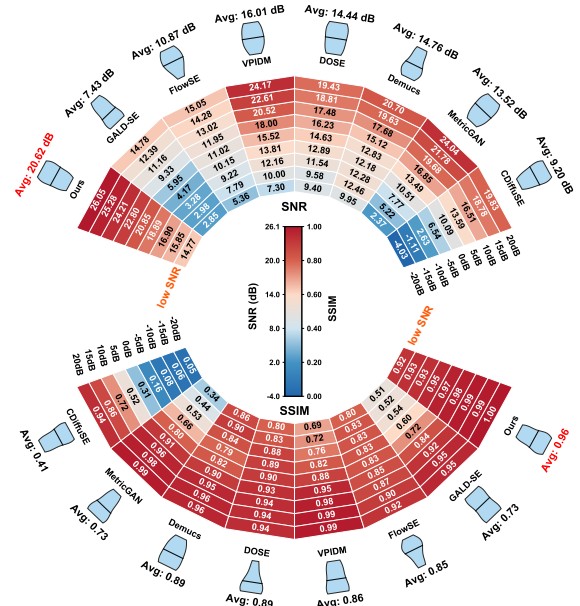

*Figure 4.* **Rydberg 4-bins EM dataset.** SNR (top) and SSIM (bottom) versus input SNR for the FracKGMF.

**EARS-WHAM!.** A more realistic and challenging speech benchmark (Richter et al., 2024) mixing anechoic speech with real environmental noise (WHAM! (Wichern et al., 2019)), featuring stronger non-stationarity and broader variability. Mixtures use SNRs uniformly sampled from $[-2.5, 17.5]$ dB with loudness normalization (16 kHz).

### 5.2. Performance on Atomic Antenna Signals

Figure 4 reports the enhancement performance on the Rydberg 4-bins EM dataset in terms of SNR (top) and SSIM (bottom) across a wide range of input conditions from $-20$ dB to 20 dB. At -20 dB input SNR, FracKGMF yields a mean SNR improvement of 34.77 dB across the test set. Across three EM datasets, FracKGMF delivers an average SNR improvement of 33.04 dB under the extreme -20 dB input condition. Overall, our method achieves the best robustness under severe corruption, obtaining the highest average SNR improvement (Avg: 20.62 dB) and the highest structural fidelity (Avg: 0.96 SSIM), outperforming all competing baselines by a clear margin. In particular, the advantage is most pronounced in the low SNR regime (e.g., $-20$ dB to 0 dB), where existing methods exhibit either insufficient noise suppression (lower SNR) or degraded structural preservation (lower SSIM), while our approach maintains consistently strong denoising and TF structure recovery.

Figure 5 reports output SNR and SSIM across SNR levels from -20 to 20 dB on EM datasets (20-bins). FracKGMF achieves the best SNR and SSIM on all SNR level, demonstrating stable robustness with enhanced performance under varying noise interference. Gains are largest in the most

*Table 1.* **Results on VoiceBank+DEMAND.** We compare FracKGMF with recent speech enhancement baselines. Metrics include PESQ, CSIG, CBAK, COVL, SSNR, and STOI (higher is better). "–" indicates values not reported in the original papers.

| Methods | Pub/Year | Params | PESQ ↑ | CSIG ↑ | CBAK ↑ | COVL ↑ | SSNR ↑ | STOI ↑ |
|---|---|---|---|---|---|---|---|---|
| Noisy | - | - | 1.97 | 3.3 | 2.44 | 2.63 | 1.68 | 0.91 |
| CDiffuSE | ICASSP/2022 | 3.3M | 2.52 | 3.72 | 2.91 | 3.10 | 5.28 | 0.91 |
| MetricGAN-OKDv2 | ICML/2023 | 1.9M | 3.12 | 4.17 | 3.13 | 3.64 | - | - |
| DR-DifuSE | AAAI/2023 | 3.55M | 3.09 | 4.38 | 3.57 | 3.76 | 9.52 | 0.95 |
| DOSE | NeurIPS/2023 | 2.3M | 2.56 | 3.83 | 3.27 | 3.19 | - | 0.94 |
| VPIDM | TASLP/2024 | 65.6M | 3.16 | 4.23 | 3.53 | 3.70 | - | - |
| Dual-S4D | TASLP/2024 | 10.8M | 2.55 | 3.94 | 3.00 | 3.23 | - | 0.93 |
| FlowSE | ICASSP/2025 | 65.6M | 3.06 | 4.14 | 3.54 | 3.61 | 9.77 | 0.95 |
| GALD-SE | SPL/2025 | 4.6M | 3.19 | 4.23 | 3.61 | 3.72 | - | 0.88 |
| BSDBNet(256) | AAAI/2025 | - | 3.11 | 4.33 | 3.58 | 3.73 | - | 0.95 |
| VPVID | TASLP/2025 | 65.6M | 3.09 | 4.24 | 3.58 | 3.67 | 9.94 | 0.95 |
| RestoreGrad | ICML/2025 | 3.4M | 2.51 | 3.80 | 3.00 | 3.14 | 5.92 | - |
| MFSE | IJCAI/2025 | 2.9M | 3.17 | 4.46 | 3.79 | 3.89 | **10.63** | 0.95 |
| Ours | - | 2.0M | **3.32** | **4.57** | **3.85** | **4.04** | 10.37 | **0.96** |

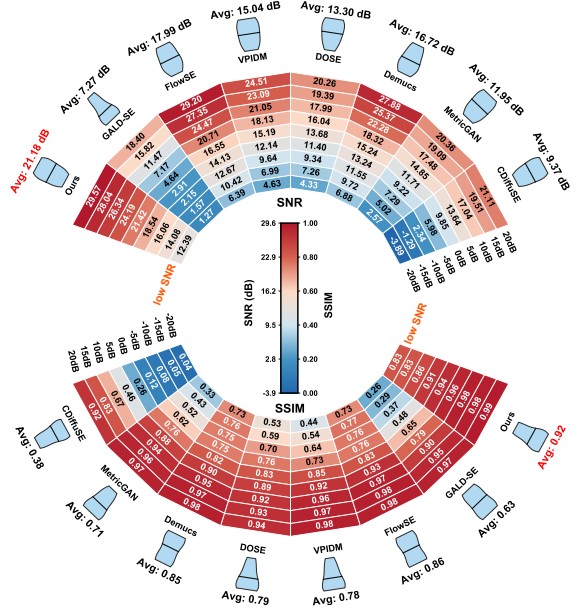

*Figure 5.* **Rydberg 20-bins EM dataset.** SNR (top) and SSIM (bottom) versus input SNR; FracKGMF yields the strongest improvements under severe corruption.

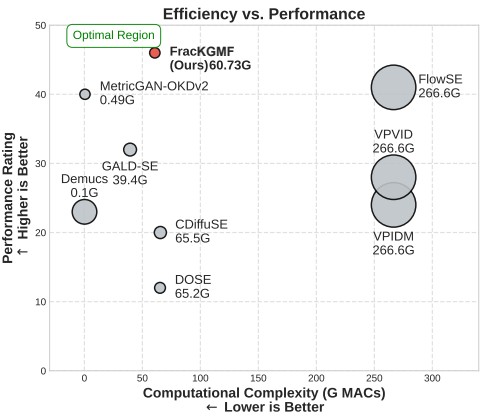

*Figure 6.* Cross Domain Generalization Efficiency Performance Trade-off. All models are trained on EARS-WHAM! and directly evaluated on VoiceBank+DEMAND (no fine tuning), to assess domain generalization. The y-axis reports the overall speech enhancement quality aggregated over six speech metrics (higher is better). The x-axis indicates computational complexity (further left is lower cost). Bubble size denotes the number of parameters (smaller bubbles indicate fewer parameters).

### 5.3. Performance on Speech Datasets

Across both VoiceBank+DEMAND and the more challenging EARS-WHAM! benchmark, our method achieves consistently strong speech enhancement quality while remaining highly efficient. On VoiceBank+DEMAND (Table 1), FracKGMF attains the best overall perceptual quality and intelligibility with 3.32 PESQ, 4.57 CSIG, 3.85 CBAK, 4.04 COVL, and 0.96 STOI using only 2.0M parameters, outperforming substantially larger models (e.g., 10.8M–65.6M parameters) as well as recent diffusion/flow-based systems. While MFSE reports a slightly higher SSNR (10.63 vs. our 10.37), our consistent gains on PESQ/CSIG/CBAK/COVL/STOI indicate improved perceptual detail preservation and intelligibility, rather than

challenging low SNR regimes: at -20 dB, FracKGMF provides an average gain of 32.39 dB, and we achieved a SNR 5.51 dB higher than the best baseline; at -5 dB, we achieve 18.54 vs 14.13 (+4.41 dB) and 0.91 vs 0.76 (+0.15). As SNR increases, performance saturates and the margin narrows as expected, while our approach remains competitive and reaches the best SSIM at 20 dB (0.99). These results align with our design: FracConv strengthens detail sensitive modeling in high coupling regions, and wiener guided fusion improves per-bin view reliability selection, yielding the strongest benefits under severe noise and distribution shifts.

*Table 2.* **Results on EARS-WHAM!.** We report PESQ, CSIG, CBAK, COVL, SSNR, and STOI (higher is better). Baseline results are obtained from their official open-source implementations under the same evaluation protocol; "–" denotes unreported values.

| Methods | PESQ ↑ | CISG ↑ | CBAK ↑ | COVL ↑ | SSNR ↑ | STOI ↑ |
|---|---|---|---|---|---|---|
| Noisy | 1.24 | 2.75 | 2.10 | 2.02 | -0.80 | 0.82 |
| CDiffuSE | 1.47±2e-2 | 2.98±2e-2 | 2.29±1e-2 | 2.27±1e-2 | -0.11±2e-2 | 0.82±1e-4 |
| DEMUCS | 1.87±2e-2 | 3.28±4e-2 | 3.03±1e-2 | 2.63±2e-2 | 7.61±5e-2 | 0.89±1e-2 |
| MetricGAN-OKDv2 | 2.05±1e-2 | 3.52±2e-2 | 2.87±1e-2 | 2.84±1e-2 | 4.31±3e-2 | 0.89±1e-2 |
| DOSE | 1.64±1e-2 | 2.50±3e-2 | 2.83±5e-2 | 2.04±2e-1 | 7.16±4e-1 | 0.87±2e-2 |
| VPIDM | 2.38±2e-2 | 3.50±1e-2 | 2.97±3e-2 | 2.91±2e-2 | 6.98±2e-1 | 0.92±1e-4 |
| GALD-SE | 2.37±1e-2 | 3.33±3e-2 | 2.98±2e-2 | 2.89±7e-2 | 7.18±1e-1 | 0.91±2e-3 |
| FlowSE | 2.33±3e-3 | 3.18±1e-2 | 3.02±1e-2 | 2.72±1e-2 | 8.35±2e-2 | 0.92±3e-4 |
| VPVID | 2.33±1e-2 | 3.33±1e-2 | 2.83±1e-2 | 2.80±1e-2 | 5.13±1e-1 | 0.86±1e-2 |
| Ours | **2.77±1e-3** | **4.19±3e-3** | **3.50±8e-4** | **3.53±2e-3** | **8.77±8e-3** | **0.94±3e-5** |

*Table 3.* **Ablations on VoiceBank+DEMAND.** Component-wise analysis of FracConv and knowledge guided fusion over 3 seeds.

| Model | PESQ↑ | CSIG↑ | CBAK↑ | COVL↑ |
|---|---|---|---|---|
| **Component ablations** | | | | |
| StdConv | 3.14±1e-2 | 4.49±1e-2 | 3.77±1e-2 | 3.89±1e-2 |
| LRF view only | 3.21±3e-2 | 4.53±3e-2 | 3.81±2e-2 | 3.95±3e-2 |
| FD view only | 3.25±1e-2 | 4.52±1e-2 | 3.80±1e-2 | 3.97±1e-2 |
| LRF+Frac (sum) | 3.25±2e-2 | 4.52±1e-3 | 3.82±1e-2 | 3.97±1e-2 |
| **FracConv floor $\varepsilon$** | | | | |
| $\varepsilon$=0.1 | 3.30±1e-2 | 4.57±1e-2 | 3.84±1e-2 | 4.03±4e-3 |
| $\varepsilon$=0.01 | 3.29±3e-3 | 4.56±2e-2 | 3.82±2e-2 | 4.02±1e-2 |
| $\varepsilon$=0.001 | 3.32±1e-2 | 4.56±1e-2 | 3.83±1e-2 | 4.03±8e-4 |
| **FracKGMF(Ours)** | 3.32±1e-2 | 4.57±1e-2 | 3.85±1e-2 | 4.04±1e-2 |

relying on aggressive suppression that primarily boosts energy-based metrics. On EARS-WHAM! (Table 2), which exhibits stronger non-stationarity and domain variability, our method achieves the best performance across all metrics (2.77 PESQ, 4.19 CSIG, 3.50 CBAK, 3.53 COVL, 8.77 SSNR, 0.94 STOI), surpassing representative convolutional (DEMUCS), GAN-based (MetricGAN-OKDv2), and diffusion/flow-based (CDiffuSE, VPIDM, FlowSE, VPVID) baselines under the same evaluation protocol. Overall, these results suggest that fractional detail modeling and KGMF jointly improve robustness under heavy noise, yielding a favorable quality efficiency trade-off across both in-domain and challenging cross domain settings.

### 5.4. Cross Domain Generalization Efficiency Performance

In Figure 6, all models are trained on EARS-WHAM! and directly evaluated on VoiceBank+DEMAND without fine tuning to measure cross domain generalization. The plot shows a clear efficiency performance trade-off: diffusion/flow methods achieve competitive quality but at very high GMACs, while lighter baselines tend to lose quality under domain shift. FracKGMF (Ours) falls in the highlighted optimal region, delivering strong aggregated performance with substantially lower compute and a compact parameter footprint. This indicates a more favorable generalization-aware Pareto frontier for practical deployment.

### 5.5. Ablation analysis.

Table 3 validates the effectiveness of each component. Replacing standard convolutions with either the LRF view or the FD view consistently improves perceptual metrics, indicating that both conservative aggregation and detail-sensitive modeling contribute beyond a single view backbone. Notably, the FD view achieves strong gains in PESQ/COVL, but still lagged behind FracKGMF, suggesting that recovering fine TF details alone can be insufficient under interference dominated observations. Directly summing the two views (naive fusion) provides limited improvement, implying that multi-view complementarity does not automatically translate into robust enhancement. In contrast, FracKGMF yields the best overall trade-off, improving PESQ/CSIG/CBAK/COVL and SSNR with low variance across seeds, which supports the role of reliability aware fusion for preventing brittle view dominance at extremely low SNR. Finally, varying the FracConv stabilization constant $\varepsilon$ over three orders of magnitude causes only minor performance changes, demonstrating that the proposed fractional detail modeling is not overly sensitive to a narrow hyperparameter choice.

## 6. Conclusion

We present FracKGMF, a robust enhancement framework designed for extremely low SNR. It couples a large receptive field (LRF) view for efficient long context coverage with a FracConv based fractional detail (FD) view that shapes distance decay interactions to preserve fine structure under strong noise and signal coupling. A wiener derived reliability reference further enables knowledge guided multi-view fusion (KGMF) to calibrate per-bin view weighting, preventing excessive suppression in uncertain regions while retaining aggressive restoration when evidence is reliable. Our analysis provides approximation and stability guaranties for the fractional envelopes and establishes reliability calibrated fusion behavior, and experiments on speech and EM benchmarks with realistic reception noise show consistent gains, especially in the most adverse low SNR conditions.

## Acknowledgements

This work was supported by the Major Program of National Natural Science Foundation of China (T2495251), the Key Program of the National Natural Science Foundation of China (62136005) and the Special Fund for Science and Technology Innovation Teams of Shanxi Province (No.202304051001001).

## Impact Statement

This paper presents work whose goal is to advance the field of Machine Learning. There are many potential societal consequences of our work, none of which we feel must be specifically highlighted here.

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

## Appendix

In the supplemental material:

## A. Additional Theory and Proofs

This appendix provides detailed proofs for the theoretical claims in Sec. 4, whose goal is to justify the core architectural choices of FracConv and KGMF from three complementary angles: expressivity, stability, and risk controlled fusion. First, we formalize the expressivity of FracConv's distance decay modeling by showing that a simplex mixture of a small number of fractional envelopes can uniformly approximate continuous density weighted normalized decay profiles over the finite kernel footprint $\Omega_{\mathbf{k}}$ (Thm. 4.1). The proof constructs a midpoint discretization over $\alpha \in [\alpha_{\min}, \alpha_{\max}]$ and controls the discretization error through the uniform sensitivity of the normalized envelope with respect to $\alpha$, quantified by $\sup_{\alpha,u} |\partial_\alpha W_\alpha(u)|$ (Lemma A.1), yielding an $O(1/M)$ approximation rate. Second, we establish stability guarantees for multi-index mixing in FracConv (Prop. 4.2 and Cor. 4.3). Since each $W_{\alpha_m}$ is bounded on $\Omega_{\mathbf{k}}$ and the mixing weights lie in the simplex, the mixed envelope remains uniformly bounded. This yields an $\ell_1$-controlled operator gain bound relative to the base kernel and implies Lipschitz continuity of the induced convolution operator with respect to $\| \cdot \|_1$ perturbations of the mixing weights. Third, we provide a risk control interpretation of KGMF as reliability calibrated averaging between the LRF and FD estimators. The two-view softmax reduces to a logistic selector $\widehat{\lambda}(Z) = \sigma(Z)$ with Lipschitz constant at most $\frac{1}{4}$ (Lemma 4.4). Under the stated first order condition, if the implemented weight tracks the conditional risk minimizing reference weight $\lambda^\star(Z)$ within error $\varepsilon_\lambda$, then the excess MSE is bounded by an $\varepsilon_\lambda^2$-dependent term involving the squared disagreement between the two views (Prop. 4.5). As a consequence, the conditional oracle fusion is no worse than either single view, while the implemented KGMF approaches this oracle behavior when its reliability guided weight remains close to $\lambda^\star(Z)$ (Cor. 4.6). Collectively, these results explain why FracConv yields a structured yet expressive fractional detail

view while preserving operator stability, and why KGMF can leverage a physically grounded reliability reference to obtain calibrated, low-regret fusion in enhancement.

| Symbol | Meaning |
|---|---|
| $y \in \mathbb{R}^L$ | Noisy or corrupted waveform, e.g., speech or EM signal, with length $L$ |
| $s \in \mathbb{R}^L$ | Clean waveform, used as the ground-truth target |
| $\mathcal{S}(\cdot), \mathcal{S}^{-1}(\cdot)$ | STFT and inverse STFT operators |
| $Y = \mathcal{S}(y) \in \mathbb{C}^{F \times T}$ | Noisy complex STFT, where $F$ and $T$ denote frequency bins and time frames |
| $S = \mathcal{S}(s) \in \mathbb{C}^{F \times T}$ | Clean complex STFT target |
| $\widetilde{Y}, \widetilde{S} \in \mathbb{R}^{2 \times F \times T}$ | Real-form STFT representations, e.g., $\widetilde{Y} = [\Re(Y), \Im(Y)]$ and $\widetilde{S} = [\Re(S), \Im(S)]$ |
| $P_Y = \lvert Y \rvert^2$ | Noisy power spectrum |
| $\widehat{P}_S, \widehat{P}_N$ | Estimated signal and noise power spectra derived from $P_Y$ |
| $\epsilon_{\mathrm{w}} > 0$ | Numerical floor used in Wiener filtering and PSD estimation |
| $R[f,t] \in [0,1]$ | Wiener inspired per-bin recoverability score / reliability prior, Eq. (8) |
| $\widehat{Y} = R \odot Y$ | Wiener-style TF estimate obtained by applying the recoverability map to the noisy STFT |
| $X \in \mathbb{R}^{C_{\mathrm{in}} \times F \times T}$ | Input TF feature map to a KGMF block |
| $C$ | Feature channel dimension after alignment or projection |
| $\phi_{\mathrm{in}}(\cdot)$ | $1 \times 1$ channel projection or alignment layer; identity if no projection is needed |
| $\mathcal{T}(\cdot)$ | LRF block, i.e., the Large Receptive Field view extractor |
| $\mathcal{F}(\cdot)$ | FracConv block, i.e., the Fractional Detail view extractor |
| $Z_{\mathrm{lrf}}, Z_{\mathrm{fd}} \in \mathbb{R}^{C \times F \times T}$ | LRF and FD feature maps produced by the two parallel views |
| $\psi(\cdot)$ | Projection from the real-form Wiener estimate to the feature space, Eq. (10) |
| $W = \psi(\widehat{Y}) \in \mathbb{R}^{C \times F \times T}$ | Knowledge or prior view in the feature space |
| $d_{\mathrm{lrf}}[f,t], d_{\mathrm{fd}}[f,t]$ | Channel-mean squared distances from the two views to the knowledge view $W$, Eq. (11) |
| $Z = d_{\mathrm{fd}} - d_{\mathrm{lrf}}$ | Per-bin reliability statistic used by the KGMF selector, Eq. (25) |
| $s_{\mathrm{lrf}}, s_{\mathrm{fd}}$ | Selection scores for the two views, typically defined as $s = -d$ |
| $w_{\mathrm{lrf}}, w_{\mathrm{fd}}$ | Softmax fusion weights per TF bin, satisfying $w_{\mathrm{lrf}} + w_{\mathrm{fd}} = 1$ |
| $\widehat{\lambda}(Z)$ | KGMF fusion weight for the LRF view; for two views, $\widehat{\lambda}(Z) = w_{\mathrm{lrf}} = \sigma(Z)$ |
| $\lambda^{\star}(Z)$ | Conditional risk minimizing oracle fusion weight, Eq. (28) |
| $\varepsilon_\lambda$ | Effective mismatch between the implemented fusion weight $\widehat{\lambda}(Z)$ and the oracle weight $\lambda^{\star}(Z)$ |
| $Z_{\mathrm{mix}}$ | KGMF fused feature map, Eq. (12) |
| $\mathbf{k} = (k_h, k_w)$ | Kernel size for FracConv, given as height $\times$ width; usually assumed odd |
| $\Omega_{\mathbf{k}}$ | Kernel offset set, i.e., the local footprint of the convolution kernel |
| $u = (\Delta f, \Delta t) \in \Omega_{\mathbf{k}}$ | A kernel offset in the TF plane relative to the kernel center |
| $r(u) = \lVert u \rVert_2$ | Radial distance of offset $u$ to the kernel center |
| $c_y = (k_h - 1)/2, \ c_x = (k_w - 1)/2$ | Kernel center indices when enumerating a $k_h \times k_w$ window on the discrete grid |
| $\alpha \in [\alpha_{\min}, \alpha_{\max}]$ | Fractional decay index controlling the concentration of the distance decay envelope |
| $W_\alpha(u)$ | Normalized fractional envelope, Eq. (3) or Eq. (13) |
| $\{\alpha_m\}_{m=1}^M$ | Fixed decay-index anchors, e.g., $M = 9$ anchors in $[0.1, 0.9]$ |
| $\boldsymbol{\pi} \in \mathbb{R}^M$ | Logits for fractional-envelope mixing weights |
| $\boldsymbol{w} = \mathrm{softmax}(\boldsymbol{\pi}) \in \Delta^{M-1}$ | Simplex mixing weights over fractional decay indices |
| $\Phi_{\boldsymbol{w}}(u)$ | Mixed fractional envelope $\Phi_{\boldsymbol{w}}(u) = \sum_{m=1}^M w_m W_{\alpha_m}(u)$, Eq. (5) |
| $A_c \in \mathbb{R}^{k_h \times k_w}$ | Learnable anisotropic base kernel for depthwise channel $c$ |
| $K_c(u) = A_c(u)\Phi_{\boldsymbol{w}}(u)$ | Effective FracConv kernel obtained by base-kernel and envelope factorization, Eq. (6) |

*Table 4.* Notation used in FracKGMF. $F$ and $T$ denote STFT frequency bins and time frames, and $C$ denotes the feature channel dimension after projection.

## A.1. Approximation of FracConv distance decay envelopes

**Lemma A.1** (Explicit bound on $C_W$). *Let $W_\alpha(u)$ be defined in Eq. (13). Define $r_{\min} = \min_{u \in \Omega_{\mathbf{k}}} r(u)$ and $r_{\max} = \max_{u \in \Omega_{\mathbf{k}}} r(u)$. Then for all $\alpha \in [\alpha_{\min}, \alpha_{\max}]$ and $u \in \Omega_{\mathbf{k}}$,*

$$\partial_\alpha W_\alpha(u) = -W_\alpha(u)\Big(\log(\varepsilon + r(u)) - \sum_{v \in \Omega_{\mathbf{k}}} W_\alpha(v)\log(\varepsilon + r(v))\Big), \tag{35}$$

*and hence*

$$C_W \leq \log\left(\frac{\varepsilon + r_{\max}}{\varepsilon + r_{\min}}\right). \tag{36}$$

Boundedness of $\partial_\alpha W_\alpha(u)$. Because $\Omega_k$ is finite, $\max_{u \in \Omega_k} \log(\varepsilon + r(u)) < \infty$. Moreover, the normalization term satisfies $\sum_{v \in \Omega_k} (\varepsilon + r(v))^{-\alpha} \geq (\varepsilon + r_{\max})^{-\alpha} > 0$. Hence $\partial_\alpha W_\alpha(u)$ is uniformly bounded over $\alpha \in [\alpha_{\min}, \alpha_{\max}]$ and $u \in \Omega_k$.

Proof of Theorem 4.1

*Proof.* Let

$$\Delta = \frac{\alpha_{\max} - \alpha_{\min}}{M}, \qquad I_m = [\alpha_{\min} + (m-1)\Delta, \alpha_{\min} + m\Delta], \tag{37}$$

and

$$\alpha_m = \alpha_{\min} + \left(m - \frac{1}{2}\right)\Delta. \tag{38}$$

Define

$$w_m = \int_{I_m} q(\alpha)\, d\alpha. \tag{39}$$

Then $\sum_{m=1}^{M} w_m = 1$. For any fixed $u \in \Omega_{\mathbf{k}}$, we have

$$
\begin{aligned}
\Phi_q(u) - \Phi_{\boldsymbol{w}}(u) &= \int_{\alpha_{\min}}^{\alpha_{\max}} q(\alpha) W_\alpha(u)\, d\alpha - \sum_{m=1}^{M} \left(\int_{I_m} q(\alpha)\, d\alpha\right) W_{\alpha_m}(u) \\
&= \sum_{m=1}^{M} \int_{I_m} q(\alpha)\left(W_\alpha(u) - W_{\alpha_m}(u)\right) d\alpha.
\end{aligned}
\tag{40}
$$

Taking absolute values gives

$$\left|\Phi_q(u) - \Phi_{\boldsymbol{w}}(u)\right| \leq \sum_{m=1}^{M} \int_{I_m} q(\alpha)\left|W_\alpha(u) - W_{\alpha_m}(u)\right| d\alpha. \tag{41}$$

By the mean-value theorem and the definition of $C_W$,

$$\left|W_\alpha(u) - W_{\alpha_m}(u)\right| \leq C_W |\alpha - \alpha_m|. \tag{42}$$

Since $\alpha_m$ is the midpoint of $I_m$,

$$|\alpha - \alpha_m| \leq \frac{\Delta}{2}, \qquad \forall \alpha \in I_m. \tag{43}$$

Therefore,

$$
\begin{aligned}
\left|\Phi_q(u) - \Phi_{\boldsymbol{w}}(u)\right| &\leq C_W \sum_{m=1}^{M} \int_{I_m} q(\alpha)|\alpha - \alpha_m|\, d\alpha \\
&\leq \frac{C_W \Delta}{2} \sum_{m=1}^{M} \int_{I_m} q(\alpha)\, d\alpha \\
&= \frac{C_W \Delta}{2}.
\end{aligned}
\tag{44}
$$

Substituting

$$\Delta = \frac{\alpha_{\max} - \alpha_{\min}}{M} \tag{45}$$

yields

$$\left|\Phi_q(u) - \Phi_{\boldsymbol{w}}(u)\right| \leq \frac{C_W(\alpha_{\max} - \alpha_{\min})}{2M}. \tag{46}$$

Taking the supremum over $u \in \Omega_{\mathbf{k}}$ completes the proof. $\qquad\square$

## A.2. Proof of Proposition 4.2

*Proof.* Fix one depthwise channel and write $K_{\boldsymbol{w}}[u] = A[u]\Phi_{\boldsymbol{w}}[u]$ for $u \in \Omega_{\mathbf{k}}$. We view $X$ as zero-extended outside the TF grid, equivalently $X \in \ell_2(\mathbb{Z}^2)$.

**Uniform operator bound.** Since $K_{\boldsymbol{w}}$ has finite support on $\Omega_{\mathbf{k}}$, Young's inequality gives

$$\|K_{\boldsymbol{w}} * X\|_2 \le \|K_{\boldsymbol{w}}\|_1 \|X\|_2. \tag{47}$$

For each $m$, the normalized envelope satisfies $0 \le W_{\alpha_m}(u) \le 1$. Since $\boldsymbol{w} \in \Delta^{M-1}$, we have

$$0 \le \Phi_{\boldsymbol{w}}[u] = \sum_{m=1}^{M} w_m W_{\alpha_m}(u) \le 1. \tag{48}$$

Therefore,

$$\|K_{\boldsymbol{w}}\|_1 = \sum_{u \in \Omega_{\mathbf{k}}} |A[u]\Phi_{\boldsymbol{w}}[u]| \le \sum_{u \in \Omega_{\mathbf{k}}} |A[u]| = \|A\|_1 \le B. \tag{49}$$

Thus,

$$\|F_{\boldsymbol{w}}(X)\|_2 = \|K_{\boldsymbol{w}} * X\|_2 \le B\|X\|_2. \tag{50}$$

**Sensitivity to mixing weights.** Let

$$\Delta\Phi = \Phi_{\boldsymbol{w}} - \Phi_{\boldsymbol{w}'}, \qquad \Delta K = K_{\boldsymbol{w}} - K_{\boldsymbol{w}'}.$$

Then

$$\Delta K[u] = A[u]\Delta\Phi[u].$$

Hence,

$$\|\Delta K\|_1 \le \|A\|_1 \|\Delta\Phi\|_\infty. \tag{51}$$

For any $u \in \Omega_{\mathbf{k}}$,

$$\begin{aligned}
|\Delta\Phi[u]| &= \left| \sum_{m=1}^{M} (w_m - w'_m) W_{\alpha_m}(u) \right| \\
&\le \sum_{m=1}^{M} |w_m - w'_m| \, |W_{\alpha_m}(u)| \\
&\le \sum_{m=1}^{M} |w_m - w'_m| = \|\boldsymbol{w} - \boldsymbol{w}'\|_1.
\end{aligned} \tag{52}$$

Taking the supremum over $u$ gives

$$\|\Delta\Phi\|_\infty \le \|\boldsymbol{w} - \boldsymbol{w}'\|_1. \tag{53}$$

Therefore,

$$\|\Delta K\|_1 \le B\|\boldsymbol{w} - \boldsymbol{w}'\|_1. \tag{54}$$

Applying Young's inequality again,

$$\begin{aligned}
\|F_{\boldsymbol{w}}(X) - F_{\boldsymbol{w}'}(X)\|_2 &= \|\Delta K * X\|_2 \\
&\le \|\Delta K\|_1 \|X\|_2 \\
&\le B\|\boldsymbol{w} - \boldsymbol{w}'\|_1 \|X\|_2.
\end{aligned} \tag{55}$$

This proves the claim. $\qquad\square$

### A.3. Proof of Corollary 4.3

*Proof.* For any $u \in \Omega_{\mathbf{k}}$,

$$
\begin{aligned}
|\Phi_{\boldsymbol{w}}(u) - \Phi_{\boldsymbol{w}'}(u)| &= \left| \sum_{m=1}^{M} (w_m - w_m') W_{\alpha_m}(u) \right| \\
&\leq \sum_{m=1}^{M} |w_m - w_m'| \, |W_{\alpha_m}(u)| \\
&\leq \sum_{m=1}^{M} |w_m - w_m'| = \|\boldsymbol{w} - \boldsymbol{w}'\|_1,
\end{aligned}
\tag{56}
$$

where we used $0 \leq W_{\alpha_m}(u) \leq 1$. Taking the supremum over $u$ proves the claim. $\qquad\square$

### A.4. Proof of Lemma 4.4

*Proof.* Let $\sigma(z) = \frac{1}{1+e^{-z}}$. Then

$$
\sigma'(z) = \sigma(z)\big(1 - \sigma(z)\big).
\tag{57}
$$

Since $0 \leq \sigma(z) \leq 1$, we have

$$
|\sigma'(z)| \leq \frac{1}{4}, \qquad \forall z \in \mathbb{R}.
\tag{58}
$$

By the mean-value theorem, for any scalar $Z, Z'$,

$$
|\sigma(Z) - \sigma(Z')| \leq \sup_{\xi \in [Z, Z']} |\sigma'(\xi)| \, |Z - Z'| \leq \frac{1}{4} |Z - Z'|.
\tag{59}
$$

For tensor-valued reliability statistics, the same argument applies pointwise. $\qquad\square$

### A.5. Proof of Proposition 4.5

*Proof.* Fix $Z$ and define

$$
\Delta := \widehat{S}_{\mathrm{lrf}} - \widehat{S}_{\mathrm{fd}}, \qquad E_{\mathrm{fd}} := \widehat{S}_{\mathrm{fd}} - S.
\tag{60}
$$

For any $\lambda \in [0, 1]$, the fused estimation error can be written as

$$
\widehat{S}_\lambda - S = (1 - \lambda)\widehat{S}_{\mathrm{fd}} + \lambda \widehat{S}_{\mathrm{lrf}} - S = E_{\mathrm{fd}} + \lambda\Delta.
\tag{61}
$$

Define the conditional risk

$$
\mathcal{R}(\lambda; Z) := \mathbb{E}\Big[ \|\widehat{S}_\lambda - S\|_2^2 \,\Big|\, Z \Big] = \mathbb{E}\big[ \|E_{\mathrm{fd}} + \lambda\Delta\|_2^2 \,\big|\, Z \big].
\tag{62}
$$

Expanding the square gives

$$
\mathcal{R}(\lambda; Z) = \mathbb{E}\big[ \|E_{\mathrm{fd}}\|_2^2 \,\big|\, Z \big] + 2\lambda \mathbb{E}[\langle E_{\mathrm{fd}}, \Delta \rangle \,|\, Z] + \lambda^2 \mathbb{E}\big[ \|\Delta\|_2^2 \,\big|\, Z \big].
\tag{63}
$$

Let

$$
A(Z) := \mathbb{E}\big[ \|\Delta\|_2^2 \,\big|\, Z \big] \geq 0.
\tag{64}
$$

Under the first-order optimality condition in Eq. (29), we have

$$
\mathbb{E}\Big[ \big\langle \widehat{S}_{\lambda^\star} - S, \Delta \big\rangle \,\Big|\, Z \Big] = 0.
\tag{65}
$$

Since

$$
\widehat{S}_{\lambda^\star} - S = E_{\mathrm{fd}} + \lambda^\star \Delta,
\tag{66}
$$

this condition is equivalent to

$$
\mathbb{E}[\langle E_{\mathrm{fd}}, \Delta \rangle \,|\, Z] + \lambda^\star(Z) A(Z) = 0.
\tag{67}
$$

Substituting this relation into the quadratic expansion yields

$$\mathcal{R}(\lambda; Z) = \mathcal{R}(\lambda^\star; Z) + A(Z)\big(\lambda - \lambda^\star(Z)\big)^2. \tag{68}$$

Setting $\lambda = \widehat{\lambda}(Z)$ gives

$$\mathcal{R}(\widehat{\lambda}; Z) - \mathcal{R}(\lambda^\star; Z) = A(Z)\big(\widehat{\lambda}(Z) - \lambda^\star(Z)\big)^2. \tag{69}$$

Using the assumption

$$|\widehat{\lambda}(Z) - \lambda^\star(Z)| \le \varepsilon \quad \text{a.s.,} \tag{70}$$

we obtain

$$\mathcal{R}(\widehat{\lambda}; Z) \le \mathcal{R}(\lambda^\star; Z) + \varepsilon^2 A(Z). \tag{71}$$

Taking expectation over $Z$ yields

$$\mathbb{E}\|\widehat{S}_{\widehat{\lambda}} - S\|_2^2 \le \mathbb{E}\|\widehat{S}_{\lambda^\star} - S\|_2^2 + \varepsilon^2 \mathbb{E}\|\widehat{S}_{\text{lrf}} - \widehat{S}_{\text{fd}}\|_2^2, \tag{72}$$

which proves Eq. (31).

It remains to prove the perturbation statement. Suppose there exists an ideal reliability statistic $\widetilde{Z}$ such that

$$|\widehat{\lambda}(\widetilde{Z}) - \lambda^\star(Z)| \le \varepsilon_0 \quad \text{a.s.} \tag{73}$$

If $Z = \widetilde{Z} + \delta$ and $\widehat{\lambda}$ is $L_\lambda$-Lipschitz, then

$$|\widehat{\lambda}(Z) - \widehat{\lambda}(\widetilde{Z})| \le L_\lambda \|Z - \widetilde{Z}\| = L_\lambda \|\delta\|. \tag{74}$$

Therefore, by the triangle inequality,

$$\begin{aligned} |\widehat{\lambda}(Z) - \lambda^\star(Z)| &\le |\widehat{\lambda}(Z) - \widehat{\lambda}(\widetilde{Z})| + |\widehat{\lambda}(\widetilde{Z}) - \lambda^\star(Z)| \\ &\le L_\lambda \|\delta\| + \varepsilon_0. \end{aligned} \tag{75}$$

For the sigmoid KGMF selector $\widehat{\lambda}(Z) = \sigma(Z)$, Lemma 4.4 gives $L_\lambda \le \frac{1}{4}$. $\qquad\square$

Here $\varepsilon$ quantifies the deviation between the implemented fusion weight and the conditional risk minimizing fusion weight. This deviation may come from noise in the wiener derived reliability statistic, approximation errors in the feature mapping, or imperfect calibration of the selector. The bound therefore characterizes how the fusion risk decreases as the implemented selector better tracks the conditional oracle weight, without assuming a specific distributional form for the reliability error.

## A.6. Proof of Corollary 4.6

*Proof.* Fix $Z$. By definition, $\lambda^\star(Z)$ minimizes the conditional risk over all $\lambda \in [0, 1]$:

$$\lambda^\star(Z) = \arg \min_{\lambda \in [0,1]} \mathbb{E}\Big[\|\widehat{S}_\lambda - S\|_2^2 \,\Big|\, Z\Big]. \tag{76}$$

Since $\lambda = 1$ and $\lambda = 0$ are feasible choices, we have

$$\mathbb{E}\Big[\|\widehat{S}_{\lambda^\star} - S\|_2^2 \,\Big|\, Z\Big] \le \mathbb{E}\Big[\|\widehat{S}_{\lambda=1} - S\|_2^2 \,\Big|\, Z\Big]. \tag{77}$$

Using the fusion definition,

$$\widehat{S}_{\lambda=1} = \widehat{S}_{\text{lrf}}, \tag{78}$$

so

$$\mathbb{E}\Big[\|\widehat{S}_{\lambda^\star} - S\|_2^2 \,\Big|\, Z\Big] \le \mathbb{E}\Big[\|\widehat{S}_{\text{lrf}} - S\|_2^2 \,\Big|\, Z\Big]. \tag{79}$$

Similarly, since

$$\widehat{S}_{\lambda=0} = \widehat{S}_{\text{fd}}, \tag{80}$$

we also have

$$\mathbb{E}\Big[\|\widehat{S}_{\lambda^\star} - S\|_2^2 \,\Big|\, Z\Big] \le \mathbb{E}\Big[\|\widehat{S}_{\text{fd}} - S\|_2^2 \,\Big|\, Z\Big]. \tag{81}$$

Combining the two inequalities gives

$$\mathbb{E}\Big[\|\widehat{S}_{\lambda^\star} - S\|_2^2 \,\Big|\, Z\Big] \le \min\Big\{\mathbb{E}\Big[\|\widehat{S}_{\text{lrf}} - S\|_2^2 \,\Big|\, Z\Big], \mathbb{E}\Big[\|\widehat{S}_{\text{fd}} - S\|_2^2 \,\Big|\, Z\Big]\Big\}. \tag{82}$$

This proves Eq. (34). $\qquad\square$

**A.7. Training Objective**

Given noisy waveform $y$ and clean waveform $s$, let $Y = \mathcal{S}(y)$ and $S = \mathcal{S}(s)$ be the complex STFTs, and let $\widehat{S}_\theta$ denote the enhanced complex spectrum predicted by the network. We train by minimizing a weighted sum of spectral and time-domain losses:

$$\mathcal{L}(\theta) = \lambda_{\text{ri}} \, \mathcal{L}_{\text{ri}}(\theta) + \lambda_{\text{mag}} \, \mathcal{L}_{\text{mag}}(\theta) + \lambda_{\text{wav}} \, \mathcal{L}_{\text{wav}}(\theta), \tag{83}$$

where

$$\mathcal{L}_{\text{ri}}(\theta) = \left\| \Re(\widehat{S}_\theta) - \Re(S) \right\|_2^2 + \left\| \Im(\widehat{S}_\theta) - \Im(S) \right\|_2^2, \tag{84}$$

$$\mathcal{L}_{\text{mag}}(\theta) = \left\| |\widehat{S}_\theta| - |S| \right\|_2^2, \tag{85}$$

$$\mathcal{L}_{\text{wav}}(\theta) = \left\| \widehat{s}_\theta - s \right\|_1, \qquad \widehat{s}_\theta := \mathcal{S}^{-1}(\widehat{S}_\theta), \tag{86}$$

and $\lambda_{\text{ri}=0.1}, \lambda_{\text{mag}=0.9}, \lambda_{\text{wav}=0.2}$ are fixed weights reference (Jin et al., 2025).

# B. Training Setup

During training, all waveforms are segmented into fixed length sequences of 51,040 samples. If an utterance does not provide enough segments, the remaining portion is padded by repeating the preceding signal content. Time-frequency representations are computed with an FFT size and window length of 510 and a hop size of 160. The sampling rate is 16 kHz for speech and 48 kHz for electromagnetic (EM) signals. The model is trained with AdamW for up to 120 epochs using a batch size of 4. The initial learning rate is $1 \times 10^{-3}$ and is decayed by a factor of 0.5 every 30 epochs of training. Early stopping is applied to mitigate overfitting. All experiments are conducted on a single NVIDIA L40 GPU with 48 GB memory.

# C. Evaluation Metrics

In the speech enhancement task, we adopt standard evaluation metrics following prior works, including:

**PESQ (WB-PESQ)**: We report the wideband Perceptual Evaluation of Speech Quality (PESQ) score following ITU-T Rec. P.862.2, which predicts subjective Mean Opinion Score (MOS) on a 1–5 scale (higher is better). In our experiments, PESQ serves as a perceptual speech quality metric that correlates with human listening judgments.

**CSIG**: CSIG is a MOS predictor that quantifies speech signal distortion, producing scores in the range 1–5 (higher is better).

**CBAK**: CBAK is a MOS predictor of the intrusiveness of background noise, with scores from 1–5 (higher is better).

**COVL**: COVL is a MOS predictor of overall quality of the enhanced speech, ranging from 1–5 (higher is better).

**SSNR**: Segmental SNR (SSNR) computes SNR over short time segments and averages across segments, which better reflects local distortion patterns than a single global SNR (higher is better).

**STOI**: short time Objective Intelligibility (STOI) is an objective metric designed to predict speech intelligibility by measuring the correlation between short time temporal envelopes of the clean and enhanced speech signals. STOI outputs a score in $[0, 1]$, where higher values indicate better intelligibility.

We further assess the proposed method on the electromagnetic (EM) signal dataset using two complementary metrics:

**SNR**: We also report the Signal-to-Noise Ratio (SNR) to quantify the overall signal energy improvement after enhancement. The SNR measures the ratio between the clean signal power and the noise power, reported in decibels (dB), where a higher SNR indicates more effective noise suppression and better signal recovery.

**SSIM**: We use the Structural Similarity Index Measure (SSIM) to evaluate the structural consistency between the enhanced and clean EM signals. SSIM outputs a score in $[0, 1]$, where higher values indicate better reconstruction quality.

# D. Wiener Filtering Procedure and Reliability Estimation

**Wiener filtering and reliability estimation.** Algorithm 1 performs Wiener filtering in the short time Fourier transform (STFT) domain and serves two complementary roles in our framework: (i) producing a conservative enhancement estimate, and (ii) providing a statistically grounded reliability cue for knowledge guided fusion. Given a noisy waveform $y$, we compute

its complex STFT $Y = \mathcal{S}(y)$ and the corresponding power spectrum $P_Y = |Y|^2$. We estimate the noise power spectral density (PSD) $\widehat{P}_N$ using temporally smoothed minimum statistics over a local time window, exploiting the observation that noise-dominated regions tend to exhibit lower and more stable energy than signal-dominated regions. The signal PSD is then estimated by nonnegative spectral subtraction,

$$\widehat{P}_S = \max(P_Y - \widehat{P}_N, \epsilon_{\mathrm{w}}), \qquad \epsilon_{\mathrm{w}} > 0,$$

which enforces non-negativity and numerical stability.

Based on these estimates, we define the per-bin recoverability as

$$R[f,t] = \frac{\widehat{P}_S[f,t]}{\widehat{P}_S[f,t] + \widehat{P}_N[f,t] + \epsilon_{\mathrm{w}}}, \qquad R[f,t] \in [0,1].$$

The wiener-style TF estimate is obtained by applying this recoverability map to the noisy STFT:

$$\widehat{Y} = R \odot Y,$$

which attenuates unreliable TF bins while preserving the noisy phase. Here, $R$ is also used as the reliability cue for downstream knowledge guided fusion. In contrast to purely data driven confidence predictors, this wiener derived recoverability map is anchored by noise aware second order statistics and therefore provides a stable reference under severe corruption and distribution shifts.

**Why Wiener statistics can reduce the effective reliability mismatch.** In our risk analysis of knowledge guided fusion, the implemented fusion weight $\widehat{\lambda}(Z)$ is compared with the conditional risk minimizing weight $\lambda^\star(Z)$. We denote their effective mismatch by $\varepsilon_\lambda$ to distinguish it from the numerical floor $\epsilon_{\mathrm{w}}$. Wiener inspired statistics provide an interpretable mechanism for reducing this mismatch under mild conditions.

The per-bin recoverability in Eq. (8) follows the classical MMSE form under an additive noise model with locally stationary second order statistics, where the gain is determined by the relative magnitudes of the estimated signal and noise PSDs $(\widehat{P}_S, \widehat{P}_N)$. Although these assumptions do not hold exactly in practice, the resulting statistic is bounded and monotonic: when interference dominates a TF bin, $\widehat{P}_N[f,t]$ becomes large relative to $\widehat{P}_S[f,t]$, and $R[f,t]$ decreases smoothly toward zero; conversely, when the signal dominates, $R[f,t]$ approaches one. This behavior discourages overconfident decisions in ambiguous TF regions.

Moreover, temporal smoothing and minimum statistics track a lower envelope of the local power spectrum, which suppresses transient high energy components when estimating the noise background. This conservative noise-tracking procedure reduces the variance of the reliability estimate and prevents the fusion module from relying solely on highly fluctuating local TF evidence. From a fusion perspective, the wiener derived reliability does not need to be exact to be useful. As long as it preserves a stable coarse ordering between more reliable and less reliable TF bins, the induced fusion weights can remain close to the conditional optimum up to a bounded mismatch $\varepsilon_\lambda$. This is consistent with our risk bound, where the excess fusion error is controlled by the weight mismatch and the disagreement between complementary views.

## E. Supplemental Experiments and Visualizations

### E.1. Performance on Modulated Signal

Across SNR levels from -20 to 20 dB, our method achieves the best SSIM on all operating points and the best (or tied-best) SNR on -20 to 15 dB, indicating consistently superior structure preservation and denoising robustness (Figure 7). At 20 dB input SNR, FracKGMF yields a mean SNR improvement of 32.96 dB across the test set. The advantage is most pronounced in extremely low SNR regimes: compared with the strongest baseline (typically FlowSE), we improve SNR by +5.14 dB / +4.80 dB / +4.19 dB / +3.23 dB at -20/-15/-10/-5 dB, together with SSIM gains of +0.15 / +0.08 / +0.05 / +0.04. As SNR increases, all methods saturate and margins narrow as expected; nevertheless, we remain competitive in SNR and continue to improve SSIM (e.g., 0.97 at 20 dB).

**Algorithm 1** Wiener Filtering with Noise PSD Estimation in the STFT Domain

---

**Input:** noisy waveform $y \in \mathbb{R}^L$; STFT operators $\mathcal{S}(\cdot)$ and $\mathcal{S}^{-1}(\cdot)$
**Hyperparameters:** smoothing factor $\beta \in (0,1)$; window length $K$; floor $\epsilon_{\mathrm{w}} > 0$
**Output:** Wiener-enhanced STFT $\widehat{Y} \in \mathbb{C}^{F \times T}$; recoverability map $R \in [0,1]^{F \times T}$; enhanced waveform $\widehat{s} \in \mathbb{R}^L$
$Y \leftarrow \mathcal{S}(y)$                                                    // complex STFT
$P_Y \leftarrow |Y|^2$                                                          // noisy power spectrum
$\widehat{P}_N[:,1] \leftarrow \max(P_Y[:,1], \epsilon_{\mathrm{w}})$
$\overline{P}[:,1] \leftarrow P_Y[:,1]$
**for** $t = 2$ **to** $T$ **do**
    $\overline{P}[:,t] \leftarrow \beta\, \overline{P}[:,t-1] + (1-\beta)\, P_Y[:,t]$      // temporal smoothing
    $t_0 \leftarrow \max(1,\ t - K + 1)$
    $\widehat{P}_N[:,t] \leftarrow \min_{\tau = t_0,\ldots,t}\ \overline{P}[:,\tau]$          // minimum statistics
    $\widehat{P}_N[:,t] \leftarrow \max(\widehat{P}_N[:,t], \epsilon_{\mathrm{w}})$         // PSD floor
**end for**
$\widehat{P}_S \leftarrow \max(P_Y - \widehat{P}_N, \epsilon_{\mathrm{w}})$                 // signal PSD estimate
$R \leftarrow \widehat{P}_S / (\widehat{P}_S + \widehat{P}_N + \epsilon_{\mathrm{w}})$       // recoverability map
$\widehat{Y} \leftarrow R \odot Y$                                              // Wiener-style enhancement
$\widehat{s} \leftarrow \mathcal{S}^{-1}(\widehat{Y})$
**return** $\widehat{Y}, R, \widehat{s}$

---

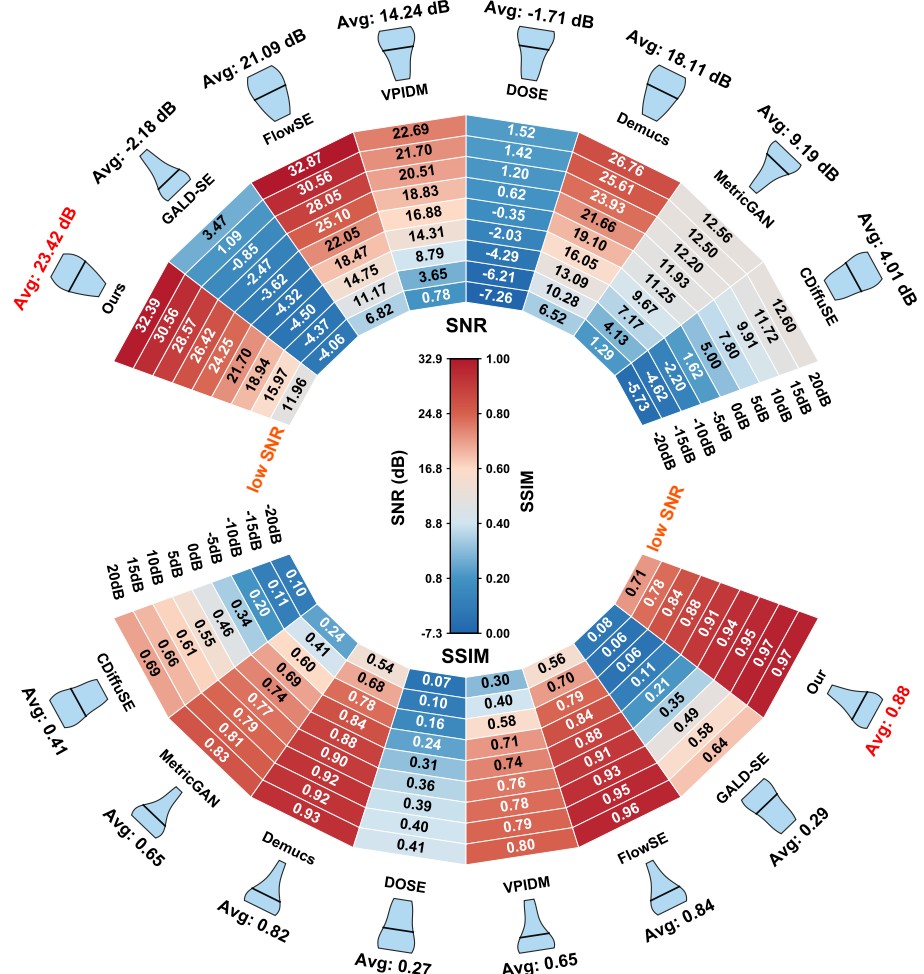

*Figure 7.* **Modulated EM dataset.** SNR (top) and SSIM (bottom) versus input SNR for the FracKGMF.

### E.2. Improvement Heatmaps on EM Benchmarks

Figures 8– 9–10 visualize the performance gaps between Ours and all baseline methods on three EM benchmarks (Rydberg 4-bins, Rydberg 20-bins, and the modulated EM dataset) via improvement heatmaps in terms of $\Delta$SNR and $\Delta$SSIM. Across all datasets and input SNR conditions, the heatmaps are dominated by positive margins (warm colors), indicating that our model consistently outperforms prior approaches under both numerical denoising gain (SNR) and structural fidelity (SSIM).

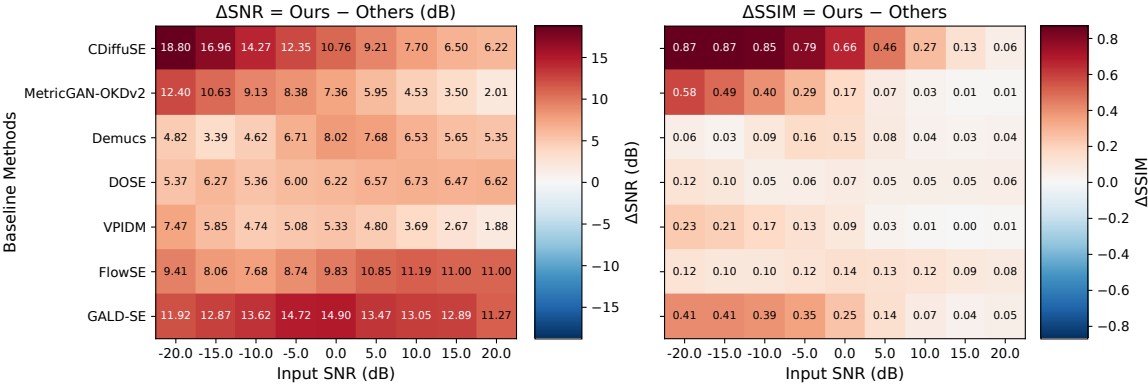

*Figure 8.* Rydberg dataset (4-bins) improvement heatmaps.

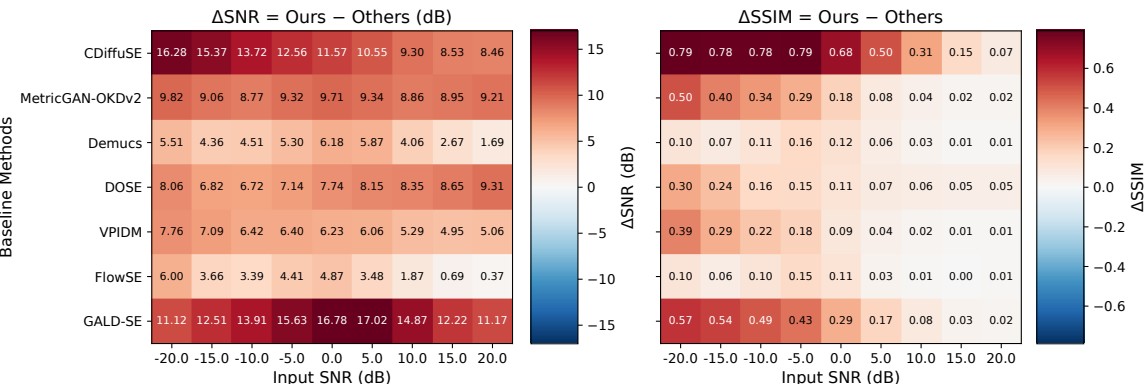

*Figure 9.* Rydberg dataset (20-bins) improvement heatmaps.

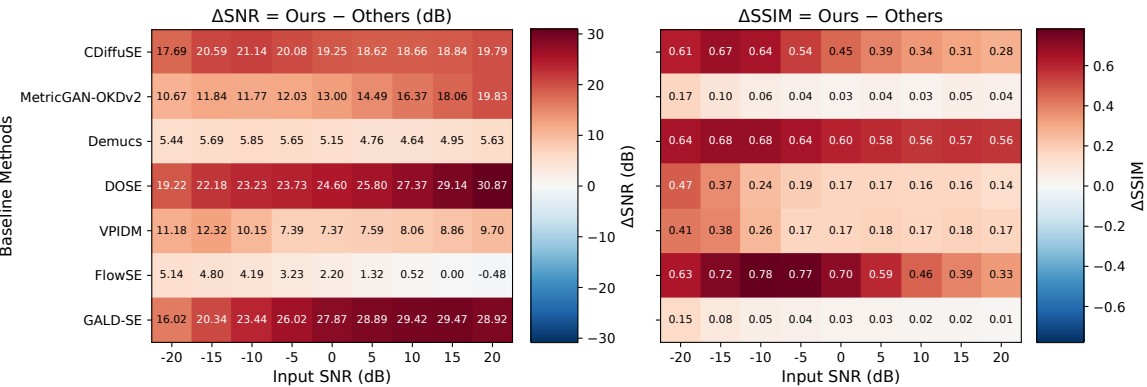

*Figure 10.* EM dataset (modulated) improvement heatmaps.

A key observation is that the improvement margins are most pronounced in the extremely low SNR regime (e.g., $-20$ dB to

$-5$ dB), where enhancement is fundamentally harder due to strong signal–noise entanglement. In this regime, our method yields large and stable $\Delta$SNR gains over diffusion-, GAN-, and flow-based baselines, while simultaneously providing substantial $\Delta$SSIM improvements, demonstrating that the recovered EM components are not only stronger in energy but also more faithful in structure.

Moreover, the consistent positive improvements across both Rydberg-style multi-bin settings (4-bins and 20-bins) and the modulated EM benchmark suggest that our approach generalizes across different spectral organizations and signal patterns, rather than overfitting to a single acquisition configuration. Overall, these heatmaps confirm that our method provides a reliable robustness advantage under distribution shifts and heavy corruption, establishing a stronger quality–robustness trade-off for EM signal enhancement.

### E.3. Effect of the fractional decay index.

Table 5 studies the impact of using a single fractional decay index $\alpha \in \{0.1, \ldots, 0.9\}$ while keeping all other components fixed. Overall, the performance exhibits a clear intermediate-index preference: $\alpha = 0.2$ achieves the best or near-best results across most perceptual and intelligibility metrics (3.26 PESQ, 4.56 CSIG, 3.81 CBAK, 4.00 COVL, 10.19 SSNR, 0.9555 STOI), suggesting that moderate decay provides the most balanced receptive-field profile for speech enhancement. In contrast, very small indices (e.g., $\alpha = 0.1$) tend to slightly underperform on energy-based metrics, while large indices (e.g., $\alpha \geq 0.8$) consistently degrade PESQ/COVL/SSNR, which is consistent with overly localized interactions that may miss long range TF context.

Importantly, the performance differences are not only reflected in the means but also in stability: intermediate indices generally yield smaller variance across runs (e.g., $\alpha = 0.2$ has $\pm 0.23$ SSNR and $\pm 0.0007$ STOI), indicating more reliable optimization and generalization behavior. These results support our design choice of adopting a multi-index fractional family: since different indices emphasize complementary context scales (global structure vs. local detail), relying on a single $\alpha$ can be suboptimal, whereas adaptive multi-index fusion can exploit the best index per TF region and improve robustness under heterogeneous noise conditions.

*Table 5.* **Effect of fractional decay index on VoiceBank+DEMAND.** We sweep a single decay index $\alpha \in \{0.1, \ldots, 0.9\}$ and report mean$\pm$std over 3 seeds for PESQ, CSIG, CBAK, COVL, SSNR, and STOI (higher is better).

| Decay indices | PESQ $\uparrow$ | CSIG $\uparrow$ | CBAK $\uparrow$ | COVL $\uparrow$ | SSNR $\uparrow$ | STOI $\uparrow$ |
|---|---|---|---|---|---|---|
| 0.1 | $3.22 \pm 0.04$ | $4.55 \pm 0.02$ | $3.78 \pm 0.03$ | $3.97 \pm 0.03$ | $9.98 \pm 0.41$ | $0.95 \pm 0.0017$ |
| 0.2 | $3.26 \pm 0.04$ | $4.56 \pm 0.01$ | $3.81 \pm 0.03$ | $4.00 \pm 0.03$ | $10.19 \pm 0.23$ | $0.96 \pm 0.0007$ |
| 0.3 | $3.23 \pm 0.03$ | $4.54 \pm 0.01$ | $3.80 \pm 0.02$ | $3.97 \pm 0.02$ | $10.16 \pm 0.12$ | $0.95 \pm 0.0012$ |
| 0.4 | $3.25 \pm 0.03$ | $4.55 \pm 0.02$ | $3.80 \pm 0.03$ | $3.99 \pm 0.02$ | $10.10 \pm 0.30$ | $0.95 \pm 0.0011$ |
| 0.5 | $3.19 \pm 0.05$ | $4.54 \pm 0.03$ | $3.73 \pm 0.04$ | $3.95 \pm 0.04$ | $9.53 \pm 0.34$ | $0.95 \pm 0.0017$ |
| 0.6 | $3.24 \pm 0.03$ | $4.56 \pm 0.01$ | $3.80 \pm 0.04$ | $3.99 \pm 0.02$ | $10.15 \pm 0.40$ | $0.95 \pm 0.0013$ |
| 0.7 | $3.25 \pm 0.03$ | $4.54 \pm 0.02$ | $3.79 \pm 0.04$ | $3.98 \pm 0.03$ | $9.96 \pm 0.43$ | $0.95 \pm 0.0012$ |
| 0.8 | $3.21 \pm 0.04$ | $4.52 \pm 0.03$ | $3.74 \pm 0.04$ | $3.95 \pm 0.03$ | $9.47 \pm 0.60$ | $0.95 \pm 0.0018$ |
| 0.9 | $3.21 \pm 0.04$ | $4.53 \pm 0.03$ | $3.75 \pm 0.04$ | $3.95 \pm 0.04$ | $9.57 \pm 0.44$ | $0.95 \pm 0.0020$ |

### E.4. Computational cost of core modules.

Table 6 reports the multiply accumulate operations (MACs) and parameter counts of the two main building blocks in our architecture. The LRFBlock contributes the dominant share of computation (956.83 MMac) while remaining parameters efficient (11.68 k), which is consistent with its role in constructing a large receptive field view that aggregates weak but globally coherent time–frequency evidence under heavy corruption. In contrast, the FracConvBlock is designed as a lightweight fractional detail view with much lower cost (78.64 MMac and 6.72 k parameters), providing localized, decay modulated interactions that emphasize sharp events and fine textures. Despite its small footprint, the fractional view is complementary rather than redundant, as it selectively recovers fast varying and high frequency details that may be smoothed by the large receptive field view, while the latter stabilizes enhancement decisions in unreliable regions. Overall, this cost breakdown supports our modular view design, in which most computation is devoted to stable global aggregation and a low overhead fractional pathway is introduced for detail recovery, leading to an improved quality efficiency trade-off without increasing the parameter budget.

| Module | MACs | Params |
|---|---|---|
| LRFBlock | 956.83 MMac | 11.68 k |
| FracConvBlock | 78.64 MMac | 6.72 k |

*Table 6.* Accumulate operations (MACs) and parameter counts of the two main building blocks in FracKGMF.

### E.5. Fractional-Index Attention Activation Analysis

Figure 11 visualizes the per index attention activations of our fractional decay family (from low to high indices), overlaid on the same intermediate TF feature map. A clear multi-scale specialization pattern emerges: low indices yield broader and smoother activation regions that emphasize global and slowly-varying structure (e.g., coherent harmonic/background components), while mid indices respond more strongly to transition bands and moderately localized regions where the TF content changes across time or frequency. In contrast, high indices produce sharper and more concentrated responses around localized salient events (e.g., narrowband peaks and abrupt bursts), indicating that they act as detectors focused on fine details with stronger locality and selectivity. This systematic redistribution of envelope mass from diffuse to concentrated interactions indicates that different fractional indices realize complementary local interaction profiles within the same TF neighborhood. This also provides the basis for our adaptive multi-index fusion in the score view: depending on the input, the model can combine broader, weaker cues (lower indices) with finer, detail sensitive interactions (higher indices) to optimize local structure. Crucially, these fractional profiles are not intended to replace long context reasoning: the LRF view provides a stable, large receptive field representation that aggregates globally consistent evidence under severe corruption, while the multi-index fractional branch injects controllable locality for detail recovery. By fusing the LRF view with the adaptively mixed fractional view under reliability calibration, the model avoids the common failure mode of a single fixed scale, which either over-smoothing fine structures or making overconfident, unstable suppression decisions in ambiguous TF regions.

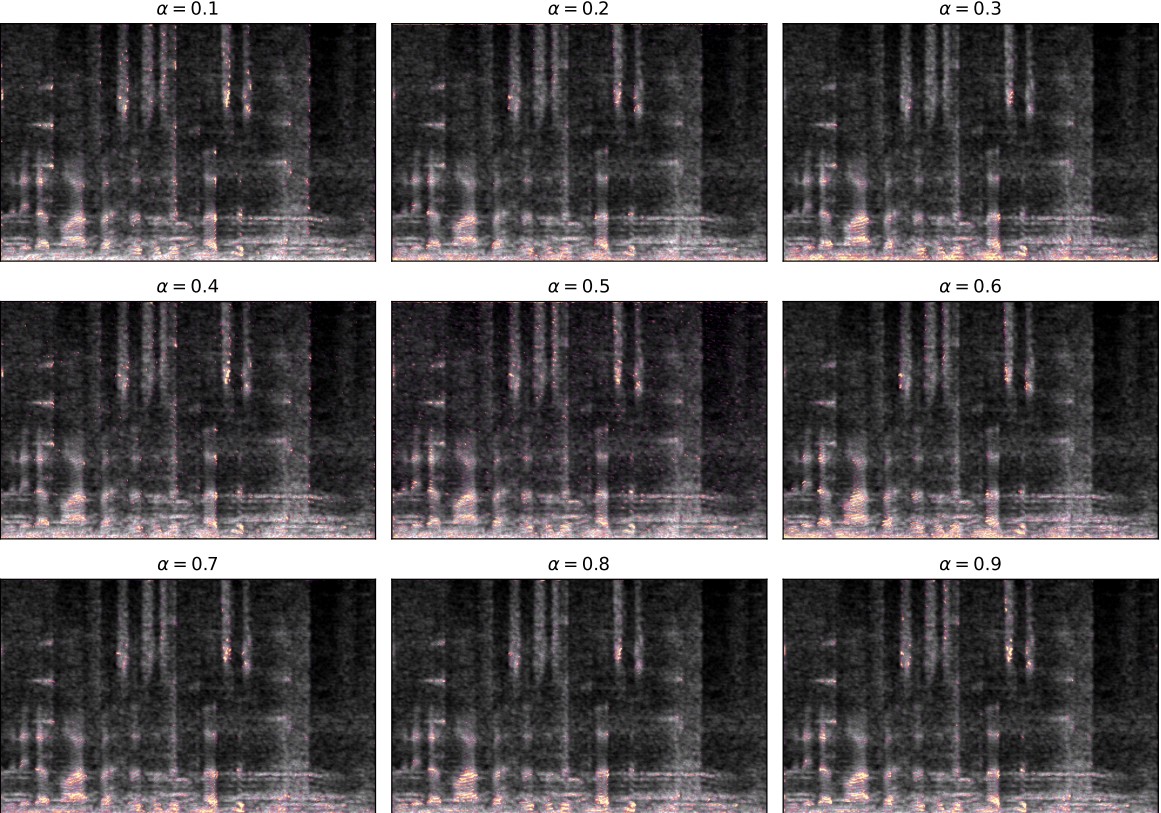

*Figure 11.* Activation visualization across fractional decay indices. Low indices emphasize global structure; mid indices emphasize transitions; high indices focus on localized sharp events, supporting adaptive multi-index fusion.

### E.6. Qualitative Comparison on EARS-WHAM!

Figure 12 provides a qualitative comparison of spectrogram enhancement results on EARS-WHAM!, covering representative baselines including diffusion/flow-based and GAN-based approaches. Overall, FracKGMF produces spectrograms that are visually closer to the clean reference, achieving a better balance between noise suppression and structure preservation.

In the highlighted regions (colored boxes), FracKGMF more faithfully restores fine-grained harmonic textures and coherent TF patterns. Specifically, in the **red boxes** (left-side regions), several baselines either retain residual background interference (insufficient suppression) or introduce over-smoothed artifacts that weaken low-frequency structures, whereas FracKGMF better recovers continuous energy trajectories with reduced noise leakage. In the **yellow box** (mid-frequency band), FracKGMF preserves clearer vertical transient components and avoids the fragmented or smeared patterns observed in competing methods, indicating improved reconstruction of non-stationary events. Similarly, in the **orange box** (right side area), FracKGMF does not exhibit artifacts, while other methods show obvious artifact patterns (e.g., false stripes and distorted textures), indicating that our method does not generate artifacts due to oversimulation of labels.

These qualitative observations align with our quantitative gains on EARS-WHAM! and further support the motivation of KGMF: by calibrating multi-view aggregation with a knowledge guided reliability cue, FracKGMF effectively integrates long range contextual evidence with fractional detail refinement, thereby avoiding both excessive suppression (detail loss) and under-suppression (residual noise) while also preventing the spurious artifacts (e.g., stripe-like distortions) that appear in competing methods under severe domain variability, leading to more natural and stable enhancement results.

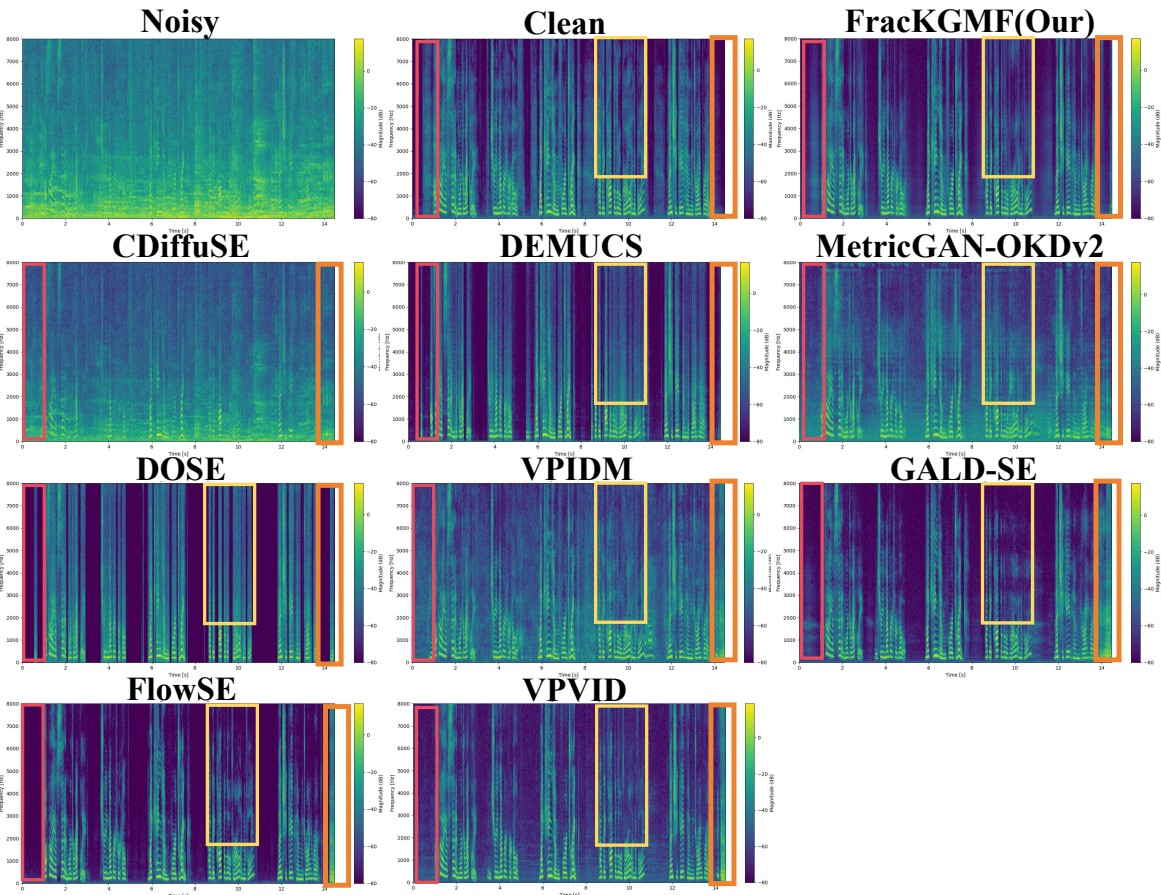

*Figure 12.* Visualized the enhancement results on the EARS-WHAM! dataset

### E.7. Qualitative Comparison on Rydberg 20-bins Dataset at -20dB

Figure 13 visualizes the enhancement results on the 20-bins EM benchmark under an extremely challenging setting $-20$ dB setting, comparing FracKGMF with representative diffusion, GAN, and convolution based baselines. As shown in the

overlaid waveforms (orange: clean; blue: enhanced), FracKGMF produces the closest match to the clean signal trajectory, recovering both the global envelope and local oscillatory patterns with minimal distortion.

In contrast, competing methods exhibit clear failure modes under such severe corruption: **CDiffuSE** introduces strong spurious fluctuations and deviates substantially from the clean waveform, **VPIDM** preserves the rough oscillation trend but suffers from noticeable amplitude mismatch and residual interference, and **Demucs/MetricGAN** tend to leave heavy residual noise, resulting in noisy plateaus and weakened structure. Moreover, methods such as **DOSE**, **FlowSE**, and **GALD-SE** show partial recovery but still produce over-smoothing or local breakdowns (e.g., unstable segments and missing sharp transitions), indicating limited robustness under extreme low SNR conditions.

Overall, this qualitative comparison demonstrates that **FracKGMF** achieves more faithful EM signal reconstruction at $-20\,\mathrm{dB}$ by suppressing noise while preserving essential waveform structure, supporting its superior robustness advantage on the multi-bin EM enhancement task.

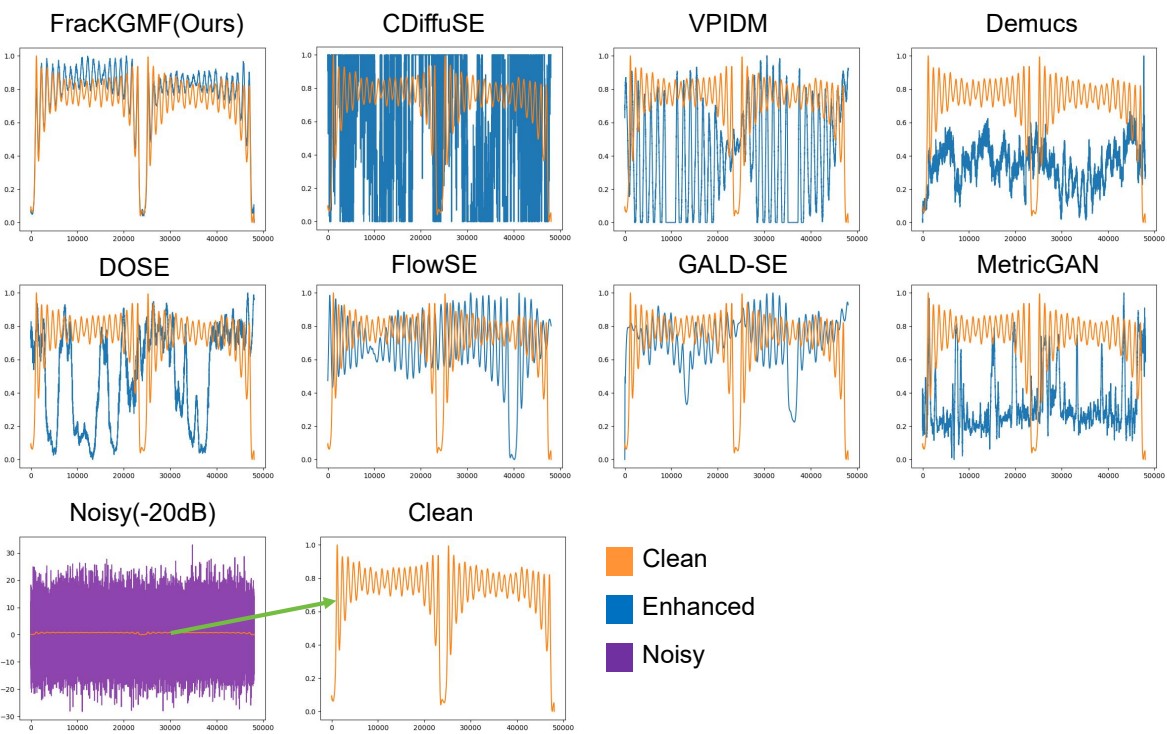

*Figure 13.* Visualized the enhancement results on the Rydberg 20-bins dataset on -20dB

