# OpenReview forum: "Robust Signal Enhancement via Fractional Detail Views and Knowledge Guided Multi-view Fusion"
_ICML.cc/2026/Conference — ICML 2026 regular_

### Official Review · Reviewer_PHCj · 2026-03-10

**Soundness:** 2
**Presentation:** 2
**Significance:** 3
**Originality:** 2
**Overall Recommendation:** 4
**Confidence:** 3

**Summary:**

This paper proposes a FracKGMF methods for signal enhancement at extremely low SNR condition. The key contributions include a Fractional Distance Decay Convolution (FracConv) and a Knowledge Guided Multi-view Fusion (KGMF). Experimental results shows that the proposed FracKGMF method acheives improved performance compared to other methods.

**Compliance With Llm Reviewing Policy:**

Affirmed.

**Final Justification:**

During rebuttal, the authors have provided further results to show that of the proposed FracConv outperforms many existing strategies including Large kernel, multi-scale conv. Moreover, further analysis shows that the learned coefficients $\alpha$ can lead to performance improvemance. Therfore, I would like to increase my rating.

**Key Questions For Authors:**

1. I'm curious about how the Wiener filter, which provides a conservative reliability estimate based on statistical principles, uses its prior-dependent hypothesis to compute the confidence of two views. Specifically, how does it avoid over-reliance on a single perspective in interference-dominant regions, while still permitting aggressive recovery when confidence is high?
2. Whether it is possible to learn the alpha rather than relying on predefined a set of alpha and use of MoE style?
3. How important is the LRFBlock since it is a large receptive field convolution? Is there any other options?

**Limitations:**

The authors should discuss more on the workings of FracConv.

**Strengths And Weaknesses:**

Pros:
1. This paper proposes a interesting FracConv which dynamically capture the TF details.
2. In general, the overall architecture of FracKGMF is simple, and the key module is lightly used in the down level and the up level.
3. This paper contains an ablation study to show the effectiveness of different components.
4. This paper provides an theoretical analysis on the expressiveness and stability of FracConv.

Cons:
1. The manuscript lacks an indepth analysis on the FracConv to show its workings in the main paper.
2. FracConv uses a mixture of exper style to obtain the final kernel, while there lacks an analysis of the distribution of the mixing coefficients. From the supplementary material E.6, it is not easy to see notable difference between different activation visualization maps of different alpha.
3. The LRF view constitutes a large proportion of parameter coefficinets and complexity, however, its functionality is not well motivated. It should also validate the possibly complementary perspective of LRF and FD.

---

> ### Author Rebuttal · Authors · 2026-03-30
>
> We sincerely appreciate your thoughtful feedback; if our response helps clarify our contributions and address your concerns, we would greatly value your reconsideration.
>
> W1:
>
> We respectfully clarify that our manuscript does provide an analysis of FracConv.
> The construction of multiple fractional decay profiles is illustrated in Fig.2–3, while its expressivity and stability are theoretically analyzed in Sec. 4.1–4.2.
> Furthermore, we have supplemented this with comparative experiments against various convolutional algorithms, which show that FracConv consistently outperforms them.
> Intuitively, FracConv can be viewed as a set of lenses with different focusing behaviors: each α defines a distance-decay profile controlling interaction range, where smaller α captures long-range dependencies and larger α emphasizes local details.
> The learned mixing weights act as adaptive tuning coefficients, dynamically selecting and combining these “lenses” based on the underlying signal–noise coupling in each region, enabling the model to flexibly resolve heterogeneous structures under diverse and interference-dominated conditions.
>
> |Model|PESQ|CSIG|CBAK|COVL|SSNR|STOI|
> |-|-|-|-|-|-|-|
> |Gaussian Conv|3.23|4.52|3.81|3.96|10.36|0.95|
> |Large Kernel(LK)(5×5)|3.23|4.51|3.78|3.96|9.91|0.95|
> |LK(7×7)|3.19|4.53|3.78|3.94|10.26|0.95|
> |LK(9×9)|3.16|4.49|3.72|3.90|9.56|0.95|
> |Multi-scale Conv|3.23|4.46|3.77|3.93|9.88|0.96|
> |**Ours**|**3.32**|**4.57**|**3.85**|**4.04**|**10.37**|**0.96**|
>
> W2 & Q2:
>
> We provide the distribution of learned mixing weights across five datasets in the anonymous link(**https://anonymous.4open.science/r/Anonymous-PHCj/**), showing non-uniform, dataset-dependent patterns rather than trivial collapse. For instance, certain α values are emphasized under stronger noise–signal entanglement, while others remain more balanced, indicating adaptive interaction selection.  Although differences across α in Supplementary E.6 may appear subtle, in TF enhancement even small spectral variations can lead to perceptually significant differences, reflecting distinct fine-grained interaction patterns. We will further improve clarity by adding guided annotations and quantitative measures. In addition, we explored learning α directly instead of using a predefined set, but found that this leads to inferior stability and no performance gain; continuous α learning tends to introduce optimization instability and less consistent interaction patterns, while MoE-style selection may cause certain α components to collapse or be under-utilized. In contrast, our discrete α design with simplex mixing ensures stable coverage of diverse interaction scales and achieves more reliable performance in practice.
>
> |Model|PESQ|CSIG|CBAK|COVL|SSNR|STOI|
> |-|-|-|-|-|-|-|
> |Continuous learning α|3.26|4.56|3.81|4.00|10.19|0.95|
> |Ours|3.32|4.57|3.85|4.04|10.37|0.96|
>
> W3 & Q3:
>
> We thank the reviewer for the insightful comment. The LRF view is implemented using a stack of lightweight residual depthwise dilated convolutions, which is a well-established and effective design for capturing large receptive fields while maintaining a favorable balance between computational efficiency and performance, rather than relying on heavier global operators. Its role is to provide a stable, low-variance contextual representation that aggregates weak but spatially consistent cues, which is particularly important under severe noise–signal entanglement . To validate its complementarity with the FD view, we include ablation results in Table 3 comparing LRF-only, FD-only, and the full model; importantly, these ablations are not constructed by simply removing one branch, but by replacing it with the same type of view, ensuring a fair comparison. The results show that each view alone is insufficient, while their combination consistently yields superior performance, demonstrating that LRF and FD capture complementary aspects of the signal and are both necessary for robust enhancement.
>
> Q1:
>
> The Wiener filter provides a statistically grounded estimate of signal recoverability by modeling the local signal-to-noise ratio, effectively reflecting how reliable each time–frequency bin is under noise–signal coupling. In our design, it is not used as a hard selector but as a soft reliability prior: we measure the consistency of each view with the Wiener estimate to derive per-bin confidence scores, which are then converted into fusion weights via a softmax function. This formulation avoids over-reliance on a single view, as interference-dominant regions (low SNR) yield low and similar confidence for both views, resulting in balanced and conservative fusion, while high-confidence regions allow the more reliable view to dominate, enabling more aggressive recovery. In this way, the Wiener-guided mechanism acts as an uncertainty-aware calibration strategy that stabilizes fusion while remaining adaptive, providing a physically grounded alternative to unconstrained attention or learned gating.

---

> > ### Author Rebuttal · Reviewer_PHCj · 2026-04-01
> >
> > Thank you for the detailed and thoughtful rebuttal.
> >
> > I appreciate the additional clarifications on the workings of the FracConv, the distribution of the mixing coeffficient, and the functionality of the Wiener filter.
> >
> > I have additional question about the distribution of $\alpha$. From the distribution of $\alpha$ shown in https://anonymous.4open.science/r/Anonymous-PHCj/, most of the 9 coefficients are relatively evenly distributed within the range of [0.10, 0.12]. This indicates that all the radial envelopes are activated, rather than selectively activated.I was wondering how the results woudl be if we set $\alpha=1/M$.  Moreover, from the additional results on the continuous learning $\alpha$, the gap between the continous learning single $\alpha$ and FracConv is relatively small.

---

> > > ### Author Response · Authors · 2026-04-02
> > >
> > > We thank you for this insightful follow-up.
> > >
> > > First, we clarify that our previous comparison was conducted against a multi-branch continuous-α variant with the same number of components (9) as FracConv, rather than a single α; we further supplemented experiments with a single continuous α, which still underperforms our method, indicating that the advantage of FracConv comes from adaptive mixing over a set of fractional decay profiles rather than a single learned decay.
> > >
> > > Second, regarding the seemingly uniform coefficient distribution, while the global average appears within a relatively narrow range (e.g., [0.088, 0.156]), the ratio between the largest and smallest coefficients reaches approximately 1.5×–2×, which is non-negligible. More importantly, this distribution is obtained after averaging over samples and time–frequency regions, and therefore masks input-dependent selectivity. In practice, different regions favor different fractional decay profiles, and the apparent near-uniformity should be understood as a marginalization effect rather than evidence of non-selective activation.
> > >
> > > In response to your suggestion, "I was wondering how the results would be if we set $\alpha=1/M$," we further evaluated the fixed uniform mixing baseline (Average mixing coefficient (1/M)) and found that it consistently performed worse than FracConv on both datasets. Importantly, on the larger and more challenging **EARS-WHAM!** dataset, the gap becomes more pronounced (e.g., **PESQ 2.55 → 2.77, SSNR 7.77 → 8.77**), while on the smaller **VoiceBank+DEMAND** dataset the difference is less significant, which is expected due to its lower diversity and weaker signal–noise heterogeneity. This observation is consistent with our design motivation: FracConv is intended to model heterogeneous time–frequency interactions, where different regions require different effective interaction ranges; in simpler settings, a single or uniformly mixed decay profile can already approximate the optimal solution reasonably well, whereas in more complex and interference-dominated conditions, adaptive mixing becomes crucial for capturing diverse structures.
> > >
> > > To further validate this behavior, we analyze the learned mixing coefficients by measuring their deviation from the uniform baseline (1/M). The results show that the coefficients do not collapse to uniform weights, but instead exhibit structured, dataset-dependent patterns. For example, **EARS-WHAM!** shows clear positive deviations at larger α, indicating a preference for long-range interactions, while **EM-4bins** remains centered around zero, suggesting minimal scale bias. Other datasets, such as **VoiceBank+DEMAND** and **EM-20bins**, exhibit moderate and multi-peak deviations, reflecting more diverse interaction requirements. These consistent yet distinct patterns demonstrate that the coefficients are not arbitrary, but adaptively select appropriate interaction scales based on data characteristics, providing strong empirical evidence that the FracConv design is both necessary and functionally meaningful.
> > >
> > > **VoiceBank+DEMAND Dataset**
> > >
> > > |Model|PESQ|CSIG|CBAK|COVL|SSNR|STOI|
> > > |-|-|-|-|-|-|-|
> > > |Noisy|1.97|3.3|2.44|2.63|1.68|0.91|
> > > |Continuous learning α (single Conv)|3.19±5e-2|4.51±3e-2|3.74±4e-2|3.93±4e-2|9.67±4e-1|0.95±2e-3|
> > > |Average mixing coefficient (1/M)|3.22±2e-2|4.52±3e-2|3.76±2e-2|3.95±3e-2|9.73±4e-1|0.95±1e-3|
> > > |**Ours**|**3.32±7e-3**|**4.57±6e-4**|**3.85±7e-3**|**4.04±8e-4**|**10.37±2e-1**|**0.96±7e-4**|
> > >
> > > **EARS-WHAM! Dataset**
> > >
> > > |Model|PESQ|CSIG|CBAK|COVL|SSNR|STOI|
> > > |-|-|-|-|-|-|-|
> > > |Noisy|1.24|2.75|2.10|2.02|-0.80|0.82|
> > > |Continuous learning α (single Conv)|2.43±4e-2|3.95±5e-2|3.22±4e-2|3.25±4e-2|6.86±3e-1|0.91±5e-3||
> > > |Average mixing coefficient (1/M)|2.55±4e-2|4.04±3e-2|3.33±3e-2|3.35±4e-2|7.77±2e-1|0.92±3e-3||
> > > |**Ours**|**2.77±1e-3**|**4.19±3e-3**|**3.50±8e-4**|**3.53±2e-3**|**8.77±8e-3**|**0.94±3e-5**|
> > >
> > > If this clarification helps address your concern, we would greatly appreciate your consideration for a positive reassessment, and wish you continued success in your research.

---

### Official Review · Reviewer_ZVwv · 2026-03-12

**Soundness:** 3
**Presentation:** 4
**Significance:** 3
**Originality:** 3
**Overall Recommendation:** 5
**Confidence:** 4

**Summary:**

The paper first identifies the T-F corruption in low SNR as a main source of quality degradation.
The proposed framework, denoted FracKGMF, is a composite system that combines two views: one that aggregates more long-term information, and a fine-grained view, assisted by a Wiener reliability reference.
The proposed method is assessed using two speech databases. It is also applied to electromagnetic signals.

**Compliance With Llm Reviewing Policy:**

Affirmed.

**Final Justification:**

I thank the authors for their efforts and for providing additional validation to their work. The audio results are also good for low-SNR values. Importantly, the algorithm can address problems from two different domains. Based on the above, my recommendation is Accept (5).

**Key Questions For Authors:**

1. What will be the influence of reberation, as it tends to smear the frequency content?
2. Can you demonstrate the results under adverse conditions, e.g., high reverberation and below-zero SNR?
3. Do we need to retrain for different noise types?

**Limitations:**

Yes

**Strengths And Weaknesses:**

Strengths:
1. The same method is applied to two different modalities.
2. It clearly outperforms competing methods, but not by a large margin.

Weaknesses:
1. For the speech signals, the SNR conditions at the input are not extreme.
2. No audio samples (e.g., using an anonymized website) can be found.
3. Enhancement under low SNR may result in the "musical noise" phenomenon. Sonogram with this problem and related performance improvement may have been beneficial.

---

> ### Author Rebuttal · Authors · 2026-03-30
>
> We sincerely thank you for your valuable comments; if our response helps clarify the contribution and addresses your concerns, we would greatly appreciate your reconsideration.
>
> W1:
>
> Our method targets general signal enhancement, with speech as one application scenario, while its strong performance on EM datasets already demonstrates robustness under extreme low SNR conditions. For speech evaluation, we follow standard benchmarks such as VoiceBank+DEMAND and EARS-WHAM! to ensure fair and reproducible comparisons, as these datasets adopt widely used SNR ranges; notably, even SNRs below 5dB are already highly challenging, and thus typical evaluations focus on moderate regimes. To further address this concern, we construct a more adverse setting using the EARS-WHAM! pipeline, covering −20dB to 0dB, where enhancement becomes particularly difficult. In this regime, our method achieves consistent and substantial improvements across all metrics over both the Noisy input and strong baselines, indicating clearly improved perceptual quality and intelligibility. Notably, our method maintains a stable positive SSNR, indicating reliable signal recovery under extreme conditions, and its advantage becomes more pronounced as SNR decreases, aligning with our design for handling severely corrupted and strongly entangled signal–noise scenarios. Due to rebuttal time constraints, we evaluate a subset of representative strong baselines, and will further expand the evaluation in the final version.
>
> |Model|PESQ|CSIG|CBAK|COVL|SSNR|STOI|
> |-|-|-|-|-|-|-|
> |Noisy(**-20dB to 0dB**)|1.17|1.72|1.35|1.40|-8.82|0.48|
> |Demucs|1.13|1.76|2.13|1.47|0.35|0.47|
> |MetricGAN-OKDv2|1.18|2.17|1.90|1.69|-3.10|0.55|
> |VPIDM|1.14|1.40|1.27|1.17|-5.41|0.51|
> |GALD-SE|1.23|1.33|1.43|1.19|-2.61|0.55|
> |FlowSE|1.14|1.20|1.52|1.09|-0.13|0.48|
> |VPVID|1.14|1.18|1.45|1.08|-0.95|0.48|
> |**Ours**|**1.37**|**2.72**|**2.25**|**2.09**|**0.46**|**0.66**|
>
> W2 & Q2:
>
> Due to data size limitations, we are showing representative samples with extremely low SNR and high reverberation, along with their spectrograms, in an anonymous link(**https://anonymous.4open.science/r/Anonymous-ZVwv/**). We will add more audio samples to the open-source link later.
>
> W3:
>
> We agree that enhancement under very low SNR is prone to the musical noise phenomenon, typically manifested as isolated narrow-band artifacts in the time–frequency(TF) domain due to over-suppression or unstable masking. To address this, we conducted additional analysis using representative spectrogram visualizations under extreme low SNR conditions. Compared with baseline methods, our approach produces fewer isolated spectral artifacts and preserves a more continuous and structured TF pattern, indicating reduced musical-noise-like distortion. This is consistent with our quantitative improvements in COVL and STOI, suggesting that the perceptual gains are not achieved through aggressive suppression. We attribute this behavior to our design, which avoids over-confident local suppression and promotes more stable signal–noise separation. We will include the corresponding visualizations and analysis in the revised manuscript.
>
> Q1:
>
> We agree that reverberation increases the difficulty of enhancement, as it smears the TF structure and reduces local spectral contrast, thereby weakening the separability between speech and interference. To evaluate this effect, we conducted additional experiments on the reverberant dataset EARS-Reverb. The results show that our method achieves consistently strong performance under reverberant conditions, with clear improvements in overall metrics, indicating that the proposed approach remains effective even when the TF structure is significantly blurred and is not limited to purely additive noise scenarios. We attribute this robustness to our design, which does not rely solely on sharp local TF cues, but instead captures more stable and structured signal characteristics under adverse conditions. We will include these additional results and analysis in the revised manuscript to further demonstrate robustness to reverberation.
>
> |Model|PESQ|CSIG|CBAK|COVL|STOI|
> |-|-|-|-|-|-|
> |Noisy(Reverb)|1.32|3.04|1.92|2.20|0.74|
> |FlowSE|1.47|2.50|1.87|1.91|0.74|
> |VPIDM|1.51|2.23|1.79|1.80|0.70|
> |**Ours**|**2.41**|**4.08**|**2.56**|**3.29**|**0.91**|
>
> Q3:
>
> Our method does not require retraining for different noise types. In all experiments reported in the paper, the model is trained once on the full dataset containing all noise types and SNR conditions, and then directly evaluated without any noise-specific fine-tuning. This design reflects a key property of our approach: it learns a noise-agnostic representation and fusion mechanism, rather than overfitting to particular noise characteristics. Consequently, the reported results already demonstrate the model’s ability to generalize across diverse noise distributions within a unified training paradigm, avoiding the need for retraining or adaptation for each noise type.

---

> > ### Author Rebuttal · Reviewer_ZVwv · 2026-03-31
> >
> > The demo files are good for both severe noise and high reverberation.

---

> > > ### Author Response · Authors · 2026-04-02
> > >
> > > We sincerely thank you for your positive feedback and for recognizing the effectiveness of our demo under both severe noise and high reverberation conditions. We also greatly appreciate your professionalism and careful evaluation as a reviewer. We wish you continued success in your research.

---

### Official Review · Reviewer_t1di · 2026-03-13

**Soundness:** 3
**Presentation:** 3
**Significance:** 3
**Originality:** 3
**Overall Recommendation:** 4
**Confidence:** 2

**Summary:**

This study presents FracKGMF, an enhancement framework designed for extremely low-SNR scenarios. FracConv and KGFM are introduced to jointly mitigate brittle enhancement under interference-dominated observations. The proposed method is evaluated on speech and realistic EM benchmarks.

**Compliance With Llm Reviewing Policy:**

Affirmed.

**Final Justification:**

Thank the authors for their response, and my concerns are addressed.

**Key Questions For Authors:**

Refer to the weaknesses

**Limitations:**

not found

**Strengths And Weaknesses:**

>Strength

The proposed method is well motivated and supported by theoretical analysis.

The paper is well organized and easy to follow.

The proposed method achieves favorable results compared with competing algorithms.

>Weaknesses

There is an inconsistency in the competing methods listed in Tables 1 and 2. Further clarification is needed regarding this issue. In addition, it is unclear why the uncertainty values are not reported in Table 1.

Some minor issues related to capitalization are observed in the titles of Sections 3.2 and 4.3.

Figure 5 presents efficiency and performance comparisons. The reviewer notes that MetricGAN-OKDv2 achieves comparable performance with significantly lower computational overhead than the proposed method. It would be helpful to report the performance of MetricGAN-OKDv2 when scaled to a similar FLOPs level as FracKGMF.

---

> ### Author Rebuttal · Authors · 2026-03-30
>
> We thank you for the insightful comment, and if our clarification helps improve the understanding of our work, we would greatly appreciate your positive consideration.
>
> W1:
>
> We thank you for pointing out this issue and apologize for any confusion. The apparent inconsistency between Tables 1 and 2 arises from differences in evaluation protocols and data availability. Specifically, several methods listed in Table 1 are only reported on the VoiceBank+DEMAND dataset in their original papers, and their implementations or pretrained models are not publicly available; therefore, we directly report their published results for fair comparison. In contrast, all methods included in Table 2 are publicly available and widely adopted state-of-the-art approaches with strong empirical performance, which enables us to evaluate them under a unified experimental setup across multiple datasets for a consistent and controlled comparison. Regarding the uncertainty values, most prior works do not report variance or standard deviation, and such information is thus unavailable for the methods in Table 1. For Table 2, since all results are obtained under our controlled experimental pipeline, we report mean ± standard deviation over multiple runs to ensure statistical reliability. We will clarify these distinctions in the revised manuscript to improve transparency.
>
> W2:
>
> Thank you for carefully reading our work. We will correct the capitalization issues in Sections 3.2 and 4.3.
>
> W3:
>
> We thank you for the insightful suggestion. We would like to clarify that Figure 5 is intended to evaluate cross-dataset generalization ability. Specifically, all models are trained on the EARS-WHAM! dataset and directly tested on the VoiceBank+DEMAND dataset without finetuning, in order to assess robustness under distribution shift. The performance on the y-axis is measured using a Performance Rating, which aggregates rankings across six evaluation metrics: each metric is ranked among all methods, assigned a score from 9 (best) to 1 (worst), and then summed to obtain the final score. Regarding the suggestion to extend MetricGAN-OKDv2 to a comparable FLOPs level, we extended its FLOPs, and the results are shown in the table below. Our method still maintains a leading position in generalization ability across datasets on most metrics, so please have no worries. We will further clarify the purpose and evaluation protocol of Figure 5 in the revised manuscript to avoid potential misunderstandings.
>
> |Methods|MACs|Params|PESQ|CSIG|CBAK|COVL|SSNR|STOI|
> |-|-|-|-|-|-|-|-|-|
> |MetricGAN-OKDv2(Large FLOPS)|73.1G|1.9M|2.67±0.01|**3.69±0.03**|3.34±0.01|3.23±0.01|7.48±0.10|0.94±0.00|
> | **Ours**|60.73G|2.0M|**3.04±0.01**|3.55±0.02|**3.68±0.02**|**3.35±0.02**|**9.85±0.02**|**0.96**±0.00|

---

> > ### Author Rebuttal · Reviewer_t1di · 2026-04-04
> >
> > Thank you for your response.

---

> > > ### Author Response · Authors · 2026-04-05
> > >
> > > Thank you very much for your recognition of our response. If our explanation has helped you better understand this work, we sincerely hope that you can also appropriately enhance your confidence, and wish you continued success in your research work.

---

### Official Review · Reviewer_xrR4 · 2026-03-20

**Soundness:** 3
**Presentation:** 3
**Significance:** 3
**Originality:** 3
**Overall Recommendation:** 3
**Confidence:** 4

**Summary:**

This paper focuses on robust signal enhancement in extremely low SNR regimes, where noise and target signals are heavily entangled and local time-frequency evidence becomes unreliable. Standard methods relying on fixed STFT representations and data-driven convolutional biases often over-suppress ambiguous regions or generate residual artifacts. The proposed FracKGMF framework mitigates these issues by integrating two complementary branches: a large receptive field module that aggregates stable long-range contextual evidence, and a FracConv-based fractional detail branch that recovers fine-grained signal structures. The two branches are fused at each time-frequency bin via a Wiener-guided reliability mechanism. This work is supplemented with theoretical analysis and experimental validation conducted on both speech and EM datasets.

**Compliance With Llm Reviewing Policy:**

Affirmed.

**Final Justification:**

After reading the rebuttal and follow-up response, I remain at weak reject. My main concern is not simply presentation quality, but that the paper still does not fully support the breadth of claim implied by its current framing.

**Key Questions For Authors:**

See Weaknesses

**Limitations:**

Yes

**Strengths And Weaknesses:**

Strengths:
1. This paper targets a real-world problem. It does not merely pursue better metric scores, but focuses on the core challenge in extremely low SNR scenarios.
2. The method is not a disjointed collection of modules and exhibits decent interpretability.
3. The method yields competitive results.

Weaknesses:
1. While the proposed FracKGMF framework improves performance, it lacks sufficient methodological novelty for an ICML submission. The LRF branch is a lightweight residual depthwise dilated convolution for large receptive fields. FracConv fuses fixed radial decay envelopes with learnable weights. KGMF uses a Wiener prior to guide softmax fusion. Each component is reasonable and well‑engineered. The method combines existing techniques instead of presenting a new learning paradigm.
2. The theoretical section is overly elaborate and it seems forced to appear rigorous.The results include approximation bounds for discrete envelope mixing, operator stability, and a regret bound for KGMF under idealized assumptions. These results are correct but only show that the method is stable and reasonable. The theory does not address key questions. It does not explain why this fractional decay family performs better than simpler options such as Gaussian decay, large-kernel depthwise convolution, or learned gating. It also does not explain why Wiener distance is suitable for guiding fusion.
3. The paper claims robustness in extremely low SNR conditions. However, the speech-related experiments do not test this range.VoiceBank+DEMAND uses training data at 0–15 dB and test data at 2.5–17.5 dB. EARS‑WHAM! covers −2.5 to 17.5 dB. The only support for extreme low SNR performance comes from EM data, not speech data. This creates an inconsistency. The paper emphasizes noise‑signal entanglement and extreme conditions, but the speech-related experiments do not support these claims.
4. The ablation study does not answer key design questions. It shows that each component improves performance, that simple summation is inferior to KGMF, and that the value of ε is not sensitive. These results confirm the effectiveness of the components but do not explain why these specific designs are necessary. The study does not compare why a Wiener-style reference is better than an end-to-end learnable confidence map. It does not show why softmax based on prior distance is more suitable than other gating methods such as elementwise attention or learned linear combination. It also does not justify why fixed anchors in [0.1, 0.9] are better than learning a continuous α per layer, simpler multi-scale convolutions, or standard large-kernel depthwise layers.
5. The construction of Figure 5 is not clearly explained. The paper describes the y-axis as an aggregated score over six speech metrics. It does not define how the aggregation is computed. It does not specify the normalization method, the weight of each metric, or whether the aggregation favors certain methods. Different metrics have different scales and distributions. An undefined composite score makes the figure difficult to interpret.
6. The term “knowledge-guided” overstates what the module actually does. The module uses a Wiener-style reliability map from the noisy STFT. It relies on a statistical prior based on local SNR estimates. This is not external knowledge, explicit rules, or an independent knowledge source. It is a signal-processing heuristic integrated into the architecture.

---

> ### Author Rebuttal · Authors · 2026-03-30
>
> We sincerely thank you for your valuable comments; if our response helps clarify the contribution and addresses your concerns, we would greatly appreciate your reconsideration.
>
> W1 & W6:
>
> We thank you for the comment but respectfully disagree that our method is a simple combination of existing techniques. The novelty lies in a principled framework rather than isolated modules: FracConv introduces a new operator that parameterizes convolution via a family of fractional distance-decay interactions, where different α correspond to distinct interaction (focusing) modes and are adaptively fused to match diverse signal–noise coupling patterns in complex scenarios. This is fundamentally different from Gaussian kernels (fixed exponential decay with isotropic assumptions), large-kernel convolutions (uniform receptive field expansion without explicit distance-aware modulation), and multi-scale designs (discrete scale aggregation without continuous decay modeling). In contrast, the fractional decay family provides a structured and continuous mechanism to regulate how information attenuates with distance, enabling flexible modeling of both local and long-range dependencies for non-stationary signal enhancement. Intuitively, this can be viewed as a set of lenses with different focusing characteristics, where each α defines a specific interaction scale and the learned mixing weights dynamically adjust their combination, allowing the model to adaptively resolve complex signal structures under varying noise conditions. For KGMF, although the Wiener filter is classical, our key contribution lies in reinterpreting it as a reliability prior for feature-level fusion, providing a physically grounded, SNR-aware confidence signal that is fundamentally different from purely data-driven attention, gating, or learned confidence maps. This explicit reliability calibration mechanism stabilizes fusion under severe noise–signal entanglement, and subsequent experiments further demonstrate its superiority over alternative fusion strategies.
>
> W2 & W4:
>
> We thank you for the careful assessment and clarify that the theoretical section is intended to rigorously formalize the design principles of our framework, establishing that the method is well-posed, stable, and theoretically grounded, rather than to overstate its contribution. While these analyses do not directly compare against alternatives such as Gaussian decay, large-kernel(LK) convolutions, or learned gating(LG), we address these key questions through targeted ablation experiments and will revise the theory section to better connect analysis with design insights. Specifically, we compare Wiener-guided fusion with learned confidence maps(LCM) and other fusion strategies (e.g., elementwise attention(EA), learned linear combination(LLC)), showing that prior-guided fusion yields more stable and better-calibrated behavior; for FracConv, we compare fixed anchors with continuous learnable-α(CL-α)  and with multi-scale and LK convolutions, demonstrating that the proposed design achieves a better trade-off between flexibility and stability. The dataset is consistent with that used in the ablation study. These results indicate that the design is not arbitrary, but reflects a principled balance between expressiveness, inductive bias, and robustness.
>
> |Model|Params(M)|PESQ|CSIG|CBAK|COVL|SSNR|STOI|
> |-|-|-|-|-|-|-|-|
> |Gaussian|2.0|3.23|4.52|3.81|3.96|10.36|0.95|
> |LK(5×5)|6.7|3.23|4.51|3.78|3.96|9.91|0.95|
> |LK(7×7)|11.2|3.19|4.53|3.78|3.94|10.26|0.95|
> |LK(9×9)|17.1|3.16|4.49|3.72|3.90|9.56|0.95|
> |LK(11×11)|24.4|3.20|4.51|3.76|3.94|9.85|0.95|
> |LG|3.8|3.24|4.54|3.79|3.98|10.07|0.95|
> |LCM|2.3|3.24|4.55|3.80|3.98|10.25|0.96|
> |EA|2.2|3.24|4.51|3.82|3.96|10.27|0.95|
> |LLC|2.0|3.25|4.55|3.80|3.98|10.03|0.96|
> |CL-α|2.1|3.26|4.56|3.81|4.00|10.19|0.95|
> |Multi-scale|3.8|3.23|4.46|3.77|3.93|9.88|0.96|
> |**Ours**|**2.0**|**3.32**|**4.57**|**3.85**|**4.04**|**10.37**|**0.96**|
>
> W3:
>
> Our method targets general signal enhancement, with speech as one application scenario; strong results on EM datasets already demonstrate robustness under extreme conditions. For speech, we adopt standard benchmarks (VoiceBank+DEMAND, EARS-WHAM!) to ensure fair and reproducible comparisons, as they follow conventional SNR settings widely used in prior work. To further validate robustness, we additionally generate low-SNR speech data (-20 to 0 dB) using the EARS-WHAM! pipeline, where our method consistently outperforms recent SOTA approaches (**see rebuttal to reviewer ZVwv for details**).
>
> W5:
>
> The y-axis in Figure 5 is computed using a ranking-based aggregation: for each metric, methods are ranked and assigned scores accordingly, and the final score is the sum over all metrics. This avoids inconsistencies from differing metric scales and removes the need for arbitrary normalization or weighting. We will clarify this in the paper.

---

> > ### Author Rebuttal · Reviewer_xrR4 · 2026-04-02
> >
> > Thank you for the rebuttal. The additional explanations are helpful, but I still have several questions that I hope the authors can clarify further. I remain uncertain about the core methodological novelty, because the current response still leaves me with the impression that the contribution is primarily a carefully engineered integration of existing ideas rather than a clearly irreducible new modeling principle. It would help if the authors could state more explicitly what the central methodological idea is that cannot be reduced to the combination of fractional decay design, large-receptive-field aggregation, and Wiener-guided fusion. Relatedly, I am still not fully convinced about the role of the theory. The rebuttal clarifies that the theory is intended to formalize stability, boundedness, and approximation properties, but this seems different from explaining why these specific design choices are preferable to simpler alternatives. I would therefore like the authors to clarify what key methodological question the theory is meant to answer beyond showing that the proposed construction is well-posed and stable.
> >
> > I also still have some concern about the paper’s positioning around extremely low SNR speech enhancement. In the main paper, the speech benchmarks do not appear to support that claim as directly as the EM experiments do, since the speech-side evaluations seem to remain largely in moderate-SNR settings. The rebuttal mentions additional low-SNR EARS-WHAM style evidence, but it is still not fully clear to me whether the speech results now fully justify the strong framing used throughout the paper. In addition, while the new ablations are useful, I am not yet convinced that they establish the necessity of the proposed design choices rather than only their effectiveness. Some of the gaps over alternative variants appear relatively modest, so I would appreciate a clearer explanation of why these results should be interpreted as evidence that the fractional anchor design and Wiener-guided fusion are specifically well motivated, rather than simply one reasonable configuration among several. I also think the terminology around “knowledge-guided” would benefit from further clarification, since the rebuttal now makes clear that the guidance comes from a Wiener-style reliability prior derived from the noisy STFT; this makes me wonder whether the term is meant in a broader signal-prior sense rather than in the stronger sense that the phrase sometimes suggests. Finally, the clarification of Figure 5 is helpful, but I would still like to understand whether the rank-based aggregation on the y-axis could obscure the actual magnitude of metric differences, and whether the same conclusion would still hold under a more transparent aggregation scheme or by reporting the constituent metrics separately.

---

> > > ### Author Response · Authors · 2026-04-05
> > >
> > > We appreciate your thoughtful evaluation and hope this clarifies your concerns. **Anonymous link:https://anonymous.4open.science/r/Anonymous-xrR4/**
> > >
> > > We will revisit our methodology for clarity. At lowSNR, noise–signal entanglement makes local time–frequency observations unreliable, causing existing methods to either over-suppress weak structures or retain residual noise and artifacts (main paper Fig.1). The core challenge is to recover weak details while avoiding noise contamination. Our key insight is that this trade-off cannot be reliably resolved within a single representation, but requires complementary views: FracConv captures detail-sensitive structures, LRF provides a low-variance estimate, and KGMF resolves their conflict through reliability-calibrated fusion.Existing approaches lack adaptive control over interaction ranges. Standard convolutions rely on fixed local aggregation, while Gaussian convolutions enforces a predefined isotropic decay, and large-kernel or multi-scale methods treat distant information uniformly or in discrete scales. As a result, they either over-suppress weak structures or retain residual noise and artifacts under heavy corruption. To address this, FracConv adopts a distance-decay formulation where the fractional parameter α acts as focus control, enabling adaptive modulation across interaction ranges. By effectively covering a broad spectrum of interaction radii, FracConv can focus on weak structured signal details under severe noise–signal entanglement. However, enhancing detail sensitivity alone is insufficient, as it becomes unstable under heavy corruption and amplifies noise. To address this, the LRF view aggregates broader context to provide a low-variance, noise-suppressed estimate. It can be interpreted as a defocused observation that smooths unreliable local variations while preserving global consistency, achieving robust aggregation with favorable efficiency–performance trade-off. FracConv and LRF are thus complementary rather than redundant: FracConv enhances weak structures but is noise-sensitive, while LRF stabilizes the representation and suppresses noise at the cost of fine details. This behavior is evident in anonymous link (moti.pdf): FracConv preserves fine structures but retains noticeable noise, while LRF produces a clean yet over-smoothed estimate; their direct combination recovers most structures but still leaves residual noise and incomplete details. This suggests that, beyond feature extraction, the challenge is to reliably leverage complementary representations under uncertainty. Existing fusion methods (attention or gating) treat all inputs as equally learnable signals and lack an explicit notion of reliability, making them prone to over-trusting corrupted observations. In contrast, KGMF introduces a Wiener-inspired reliability prior, providing a physically grounded estimate of signal confidence based on local SNR, enabling conservative suppression of unreliable evidence while preserving confident structures.
> > >
> > > Regarding theoretical issues, our current theoretical analysis primarily establishes the validity and soundness of the proposed method, but does not yet fully articulate its distinction from simpler alternatives. We acknowledge this limitation and will further strengthen the theoretical discussion in the revised version.
> > >
> > > To address concerns about lowSNR speech, we've included sample audio files at an anonymous link for you to listen to, and added visualizations to lowSNR.pdf to aid your understanding. Furthermore, the enhancement metrics for low SNR are also presented in EARS.pdf.
> > >
> > > To address concerns about ablation experiments, since the VB is a small dataset, it can quickly reflect the effectiveness of a method, but may not be able to differentiate between methods, we retested on the larger EARS dataset in the table below.
> > >
> > > |Model|PESQ|CSIG|CBAK|COVL|SSNR|STOI|
> > > |-|-|-|-|-|-|-|
> > > |Gaussian|2.52|3.99|3.30|3.31|7.40|0.92|
> > > |LK(5×5)|2.53|4.02|3.29|3.33|7.29|0.92|
> > > |LK(7×7)|2.55|4.03|3.33|3.35|7.66|0.92|
> > > |LK(9×9)|2.59|4.05|3.37|3.39|8.00|0.92|
> > > |LK(11×11)|2.52|4.01|3.30|3.32|7.46|0.92|
> > > |LG|2.50|3.97|3.28|3.29|7.28|0.92|
> > > |LCM|2.60|4.05|3.36|3.38|7.88|0.92|
> > > |EA|2.57|4.05|3.36|3.36|8.04|0.92|
> > > |LLC|2.53|4.03|3.34|3.14|7.72|0.92|
> > > |CL-α|2.62|4.09|3.38|3.41|7.96|0.93|
> > > |Multi-scale|2.55|4.04|3.33|3.35|7.64|0.92|
> > > |LRF only|2.58|4.06|3.37|3.38|8.63|0.92|
> > > |FD only|2.63|4.10|3.40|3.42|7.67|0.93|
> > > |**Ours**|**2.77**|**4.19**|**3.50**|**3.53**|**8.77**|**0.94**|
> > >
> > > In our work, "knowledge-guided" refers to Wiener-style reliability estimates derived from noisy STFT, rather than external knowledge. This prior provides a physically-based confidence measure for fusion. We will clarify this terminology in the revised edition to avoid ambiguity.
> > >
> > > Finally, for detailed Fig.5 metrics, please refer to F5.pdf in the anonymous link.
> > >
> > > If this clarification helps address your concern, we would appreciate your positive reassessment and wish you smooth progress in your work

---

### Decision · Program_Chairs · 2026-04-30

**Decision:**

Accept (regular)

**Comment:**

This paper proposes FracKGMF, a signal enhancement framework targeting extremely low-SNR conditions. Fractional Distance Decay Convolution (FracConv) and a Knowledge-Guided Multi-view Fusion (KGMF) are proposed.

The paper received 4 reviews with mixed scores - Accept, 2 Weak Accept, and Weak Reject.
After the rebuttal, 3 reviewers converged toward acceptance as the authors provided
- audio demonstrations and additional low-SNR results.
- FracConv comparisons with alternative methods
- clarifications on the role of Wiener filter

The remaining dissent comes from the following concerns
- overstated title (robust signal enhancement) compared to the 2 experimented modalities (speech, EM)
- the claim of extreme low SNR robustness is backed by EM experiments. However, the speech benchmarks only operate on moderate SNR
- The results show the effectiveness of the proposed method but doesn't convince that the particular design choice is necessary.

After weighing these concerns against the majority view, I lean toward acceptance.
The core contributions are technically sound and well-motivated, the cross-modal applicability (speech and EM) is a genuine strength.
The authors in the rebuttal provides gains over other alternatives.